# Evaluating the large-scale hydrological cycle response within the PlioMIP2 ensemble

Zixuan Han[1,2], Qiong Zhang[2], Qiang Li[2], Ran Feng[3], Alan M. Haywood[4], Julia C. Tindall[4], Stephen J. Hunter[4], Bette L. Otto-Bliesner[5], Esther C. Brady[5], Nan Rosenbloom[5], Zhongshi Zhang[6,7], Xiangyu Li[6], Chuncheng Guo[7], Kerim H. Nisancioglu[8,9], Christian Stepanek[10], Gerrit Lohmann[10,11], Linda E. Sohl[12,13], Mark A. Chandler[12,13], Ning Tan[14,15], Gilles Ramstein[15], Michiel L. J. Baatsen[16], Anna S. von der Heydt[16], Deepak Chandan[17], W. Richard Peltier[17], Charles J. R. Williams[18,19], Daniel J. Lunt[18], Jianbo Cheng[20], Qin Wen[21], Natalie J. Burls[22]

1. College of Oceanography, Hohai University, Nanjing, China
2. Department of Physical Geography and Bolin Centre for Climate Research, Stockholm University, Stockholm, Sweden
3. Department of Geosciences, College of Liberal Arts and Sciences, University of Connecticut, CT 06269, USA
4. School of Earth and Environment, University of Leeds, Woodhouse Lane, Leeds, West Yorkshire, UK
5. Climate and Global Dynamics Laboratory, National Center for Atmospheric Research, Boulder, CO 80305, USA
6. Department of Atmospheric Science, School of Environmental Studies, China University of Geosciences, Wuhan, China
7. NORCE Norwegian Research Centre, Bjerknes Centre for Climate Research, Bergen, Norway
8. Department of Earth Science, University of Bergen, Bjerknes Centre for Climate Research, Bergen, Norway
9. Centre for Earth Evolution and Dynamics, University of Oslo, Oslo, Norway
10. Alfred Wegener Institute-Helmholtz-Zentrum für Polar und Meeresforschung, Bremerhaven, Germany
11. Institute for Environmental Physics, University of Bremen, Bremen, Germany
12. Center for Climate Systems Research, Columbia University, New York, USA
13. NASA Goddard Institute for Space Studies, New York, USA
14. Key Laboratory of Cenozoic Geology and Environment, Institute of Geology and Geophysics, Chinese Academy of Sciences, Beijing, China
15. Laboratoire des Sciences du Climat et de l'Environnement, LSCE/IPSL, CEA-CNRS-UVSQ, Université Paris-Saclay, Gif-sur-Yvette, France
16. Institute for Marine and Atmospheric research Utrecht (IMAU), Department of Physics, Utrecht University, Utrecht, The Netherlands.
17. Department of Physics, University of Toronto, Toronto, Ontario, Canada
18. School of Geographical Sciences, University of Bristol, Bristol, UK
19. NCAS-Climate, Department of Meteorology, University of Reading, Reading, UK
20. School of Environmental Science and Engineering, Yancheng Institute of Technology, Yancheng, China
21. School of Geography, Nanjing Normal University, Nanjing, 210023, China; Key Laboratory of Virtual Geographic Environment (Nanjing Normal University), Ministry of Education, Nanjing, 210023, China; Jiangsu Center for Collaborative Innovation in Geographical Information Resource Development and Application, Nanjing, 210023, China
22. Center for Ocean-Land-Atmosphere Studies, George Mason University, Fairfax, Virginia, USA

*Correspondence to*: Qiong Zhang (qiong.zhang@natgeo.su.se) and Zixuan Han (zixuan.han@hhu.edu.cn)

**Abstract.** The mid-Pliocene (~3 million years ago) is one of the most recent warm periods with high $CO_2$ concentrations in the atmosphere and resulting high temperatures and is often cited as an analog for near-term future climate change. Here, we apply a moisture budget analysis to investigate the response of the large-scale hydrological cycle at low latitudes within a

13-model ensemble from the Pliocene Model Intercomparison Project Phase 2 (PlioMIP2). The results show that increased atmospheric moisture content within the mid-Pliocene ensemble (due to the thermodynamic effect) results in wetter conditions over the deep tropics, i.e., the Pacific intertropical convergence zone (ITCZ) and the Maritime Continent, and drier conditions over the subtropics. Note that the dynamic effect plays a more important role than the thermodynamic effect in regional precipitation minus evaporation (PmE) changes (i.e., northward ITCZ shift and wetter north Indian Ocean). The thermodynamic effect is to some extent offset by a dynamic effect involving a northward shift of the Hadley circulation that dries the deep tropics and moistens the subtropics in the Northern Hemisphere (i.e., the subtropical Pacific). From the perspective of Earth's energy budget, the enhanced southward cross-equatorial atmospheric transport (0.22 PW), induced by the hemispheric asymmetries of the atmospheric energy, favors an approximately 1° northward shift of the ITCZ. The shift of the ITCZ reorganizes atmospheric circulation, favoring a northward shift of the Hadley circulation. In addition, the Walker circulation consistently shifts westward within PlioMIP2 models, leading to wetter conditions over the northern Indian Ocean. The PlioMIP2 ensemble highlights that an imbalance of interhemispheric atmospheric energy during the mid-Pliocene could have led to changes in the dynamic effect, offsetting the thermodynamic effect and hence altering mid-Pliocene hydroclimate.

## 1 Introduction

Global warming can induce regional and global anomalies in the Earth's hydrological cycle, thereby regulating the balance of global water resources (Eltahir and Bras, 1996). Many studies have indicated that pronounced climate change can occur as anthropogenic $CO_2$ rises, including an increase in surface temperature (Xie et al., 2010; Long et al., 2014), Arctic amplification (Stuecker et al., 2018; Smith et al., 2019), and impacts on animal and plant populations (Root et al., 2003). Under current global warming, both observations and model simulations suggest a tendency for the wet-regions-getting-wetter-and-dry-regions-getting-drier phenomenon (Held and Soden, 2006; Wentz et al., 2007; Chou et al., 2009; Wang et al., 2012; Li et al., 2013). That is, precipitation minus evaporation (PmE) increases (decreases) in regions of climatological convergence (divergence). Note that this phenomenon is primarily focused on the ocean. A study by Greve et al. (2014) report that only 10.8% of the global land area shows the dry gets drier and wet gets wetter pattern. These changes in the large-scale hydrological cycle could induce severe climatic disasters worldwide, leading to considerable impacts on economies, ecosystems, and agriculture (Asokan and Destouni, 2014; Bengtsson, 2014). Therefore, understanding the potential processes responsible for large-scale hydrological cycle changes in a warmer climate is of great importance.

Previous studies have suggested that the thermodynamic effect caused by increased atmospheric moisture content in a warmer climate is one of the primary contributors to a tendency of wet gets wetter and dry gets drier (Chou et al., 2009; Seager et al., 2010). This mechanism directly follows the nonlinearity of the Clausius-Clapeyron relationship, which acts to increase atmospheric moisture content over regions with the warmest surface temperatures (Allen and Ingram, 2002; Stephens and Ellis, 2008). On the other hand, large-scale atmospheric circulation can change substantially due to nonuniform

temperature changes under global warming and hence induce changes in the hydrological cycle via the so-called dynamic effect (Han et al., 2019a). The dynamic effect is relatively more complicated than the thermodynamic effect among climate

models. Seager et al. (2010) demonstrated that the dynamic component is modulated by the weakening of the Hadley circulation and Walker circulation. Increased $CO_2$ concentration could directly increase atmospheric static stability over tropical oceans, favoring a slowdown of these atmospheric overturning circulations (Vallis et al., 2015). Other studies have indicated that local Hadley circulation shifts poleward due to the decreased meridional temperature gradient in response to increased $CO_2$ concentrations (Sharmila and Walsh, 2018; Hu et al., 2018a). These circulation anomalies widen the

subtropical dry zones (Previdi and Liepert, 2007; Sun et al., 2013b). In addition, Long et al. (2016) highlighted that model uncertainty in tropical rainfall comes from the discrepancies of the atmospheric circulation anomalies among models. Thus, the spread of circulation changes in response to global warming across climate models leads to a diversity of responses in the hydrological cycle.

          Proxy data indicate that the mid-Pliocene (~3 million years ago) was one of the most recent warm periods with $CO_2$

levels similar to the current anthropogenically elevated value of 400 ppm and can be considered an analog for future climate change (Dowsett et al., 2012; Burke et al., 2018; Tierney et al., 2019). Pliocene Model Intercomparison Project Phase 1 (PlioMIP1) simulations have been used to investigate how the climate system responded to mid-Pliocene boundary conditions, including elevated atmospheric $CO_2$ concentrations. These past warm climate simulations exhibit many similarities with future climate projections. For example, one robust characteristic is increased temperature from 1.8 to 3.6℃

during the Pliocene compared with the preindustrial period (PI) (Haywood et al., 2013), with Arctic amplification in response to a significant sea-ice extent decline (Howell et al., 2016; Zheng et al., 2019). These features could have reduced the meridional surface temperature gradient, inducing weaker tropical circulation (i.e., local Hadley circulation) during the Pliocene (Sun et al., 2013a; Li et al., 2015; Corvec and Fletcher, 2017). Additionally, some studies have suggested a weakened zonal sea surface temperature (SST) gradient in the Pacific during the Pliocene (Wara et al., 2005; Scroxton et al.,

2011), which would have favored weaker Walker circulation. These features could have induced large-scale changes in Pliocene hydroclimate. Using a climate simulation that captures the warming patterns seen in Early Pliocene sea surface temperature proxies, Burls and Fedorov (2017) suggested that the dynamic process might play a key role in driving wetter subtropics due to this weaker tropical circulation during the Early Pliocene warm climate compared with the future climate.

          Although PlioMIP1 can reproduce similar patterns of the change in surface temperature to the reconstructed SST,

models cannot capture the magnitude of warming at higher latitudes. For example, Dowsett et al. (2013) indicated that the ensemble of PlioMIP1 models underestimates the warming in the North Atlantic compared with the reconstructed SST. This might be induced by the uncertainties in PlioMIP1, including the uncertainty in atmospheric $CO_2$ concentrations (Salzmann et al., 2013; Howell et al., 2016) and paleogeography and bathymetry (Otto-Bliesner et al., 2016; Feng et al., 2017). PlioMIP2 models show the closed Canadian Archipelago and Bering Strait and a reduced Greenland ice sheet relative to

PlioMIP1. For one of the PlioMIP2 models, it has been shown that updating the paleogeography to PRISM4 is the major contributor to climate differences from PlioMIP1 to PlioMIP2 (Samakinwa et al., 2020). In addition, PlioMIP2 focuses on a

specific time slice during the mid-Pliocene at approximately 3.025 Ma, which could reduce the uncertainties in reconstructions (McClymont et al., 2020). Researchers have been investigating the mid-Pliocene climate by using PlioMIP2, including Arctic warming (De Nooijer et al., 2020), Atlantic meridional overturning circulation (Zhang et al., 2021b), climate sensitivity (Haywood et al., 2020), global monsoons (Zhang et al., 2021a), and subtropical rainfall changes (Pontes et al., 2020). However, it is difficult to distinguish the relative impact of the Hadley circulation and Walker circulation on Pliocene hydrological cycling at low latitudes. Fortunately, the three-pattern decomposition of global atmospheric circulation (3P-DGAC; Hu et al., 2017, 2018b, c) method can help us to decompose atmospheric circulation into zonal (i.e., local Walker circulation) and meridional (i.e., local Hadley circulation) circulation at low latitudes. We apply this method to develop moisture budget analyses, which might provide some insight into the mechanisms of hydrological cycling during the mid-Pliocene.

This paper is in the framework of updated PlioMIP2 models to quantitatively distinguish the relative contribution from zonal and meridional circulation anomalies to hydrological cycle changes. In the following section, we first introduce the PlioMIP2 models and moisture budget decomposition. We then evaluate the simulated large-scale hydroclimate cycle response within the PlioMIP2 ensemble in section 3. Section 4 provides each moisture budget component's relative contribution to investigate the potential mechanisms driving the simulated changes in the mid-Pliocene hydrological cycle. The corresponding mechanisms are discussed in section 5. The last section contains the conclusion and discussion.

## 2 Data and analytical methods

### 2.1 Climate model simulations

In this study, we use the simulations from 13 models participating in PlioMIP2 (Table 1). All models include a preindustrial (PI) simulation and a Pliocene climate simulation. In PlioMIP2 models, the boundary conditions have been updated using the new version of the U.S. Geological Survey PRISM4 dataset (Dowsett et al., 2016; Haywood et al., 2016), including soils, lakes, land ice cover, vegetation, topography, and bathymetry. The $CO_2$ level for mid-Pliocene and PI simulations are set at 400ppmv and 280ppmv, respectively. To calculate the ensemble mean, we interpolate all data onto a common grid with a 1°×1° resolution using bilinear interpolation.


Table 1: PlioMIP2 models used in this study

| Model name | Institute | PlioMIP2 reference |
| --- | --- | --- |
| 1. CESM2 | NCAR | Feng et al. (2020) |
| 2. COSMOS | Alfred Wegener Institute | Stepanek et al. (2020) |
| 3. EC-Earth3-LR | Stockholm University | Zhang et al. (2021a) |
| 4. HadCM3 | Hadley Centre for Climate Prediction and Research/Met Office UK | Hunter et al. (2019) |
| 5. GISS-E2-1-G | NASA/GISS | Kelley et al. (2020) |
| 6. IPSL-CM6A-LR | Laboratoire des Sciences du Climat et de l'Environnement (LSCE) | Lurton et al. (2020) |
| 7. CCSM4-UofT | University of Toronto, Canada | Peltier and Vettoretti (2014) Chandan and Peltier (2017) |
| 8. NorESM1-F | NORCE Norwegian Research Centre, Bjerknes Centre for Climate Research, Bergen, Norway | Li et al. (2020) |
| 9. NorESM-L | NORCE Norwegian Research Centre, Bjerknes Centre for Climate Research, Bergen, Norway | Li et al. (2020) |
| 10. CCSM4-Utrecht | IMAU, Utrecht University | Baatsen et al. (2021), in prep |
| 11. HadGEM3 | Hadley Centre for Climate Prediction and Research/Met Office UK | Williams et al. (2021) |
| 12. CCSM4 | NCAR | Feng et al. (2020) |
| 13. CESM1.2 | NCAR | Feng et al. (2020) |

## 2.2 Development of moisture budget decomposition

To examine the changes in precipitation (P) minus evaporation (E) in the PlioMIP2 mid-Pliocene experiments relative to their respective PI simulation, we decompose the moisture budget equation based on Seager et al. (2010), i.e.,

$$\delta(\overline{P}-\overline{E}) \approx \underbrace{-\frac{1}{\rho_w g}\nabla\cdot\int_0^{p_s}(\overline{V_0}\delta\overline{q})dp}_{\delta TH}\underbrace{-\frac{1}{\rho_w g}\nabla\cdot\int_0^{p_s}(\overline{q}\,\delta\overline{V_0})dp}_{\delta MCD}+R \qquad (1)$$

Here, $g$ is gravity, $\rho_w$ is the density of water, $\vec{V}$ is the horizontal wind, and $q$ is the specific humidity. $\delta(\cdot)$ is the annual mean difference of variables between the warmer climate state (mid-Pliocene) and PI simulation. Subscript 0

represents the variables in the PI simulation. In the warmer climate, the change in P minus E [PmE, the left-hand side of Eq. (1)] is balanced by the thermodynamic ($\delta$TH, induced by increased specific humidity) and dynamic ($\delta$MCD, induced by circulation anomalies) contributions and residual term (R, which is mainly involved in the contributions from high-frequency variability of transient eddies, nonlinear effects and surface boundary terms).

Since we are interested in understanding the relative contribution from zonal circulation (i.e., local Walker

circulation) changes and meridional circulation (i.e., local Hadley circulation) anomalies to the changes in PmE in a warmer climate, we further apply the three-pattern decomposition of global atmospheric circulation (3P-DGAC; Hu et al., 2017, 2018b, c) method in this study. The horizontal, meridional, and zonal circulations that can be viewed as the global generalization of the Rossby wave in the middle-high latitudes and the Hadley and Walker circulations in the low latitudes are defined to decompose the global atmospheric circulation into a superposition of the horizontal, meridional, and zonal

circulations by using the 3P-DGAC method.

Based on the essential features of the Rossby, Hadley and Walker circulations, Hu et al. (2017) defined the 3D horizontal circulation $\vec{V}_R$, meridional circulation $\vec{V}_M$ and zonal circulation $\vec{V}_Z$ in the spherical $\sigma$-coordinate system as follows:

$$\begin{cases}\vec{V}_R(\lambda,\theta,\sigma)=u_R(\lambda,\theta,\sigma)\vec{i}+v_R(\lambda,\theta,\sigma)\vec{j},\\ \vec{V}_M(\lambda,\theta,\sigma)=v_M(\lambda,\theta,\sigma)\vec{j}+\dot{\sigma}_M(\lambda,\theta,\sigma)\vec{k},\\ \vec{V}_Z(\lambda,\theta,\sigma)=u_Z(\lambda,\theta,\sigma)\vec{i}+\dot{\sigma}_Z(\lambda,\theta,\sigma)\vec{k}.\end{cases} \qquad (2)$$

and the following continuity equations are satisfied:

$$\begin{cases} \dfrac{1}{\sin\theta}\dfrac{\partial u_R}{\partial\lambda}+\dfrac{1}{\sin\theta}\dfrac{\partial(\sin\theta v_R)}{\partial\theta}=0, \\[2mm] \dfrac{1}{\sin\theta}\dfrac{\partial(\sin\theta v_M)}{\partial\theta}+\dfrac{\partial\dot{\sigma}_M}{\partial\sigma}=0, \\[2mm] \dfrac{1}{\sin\theta}\dfrac{\partial u_Z}{\partial\lambda}+\dfrac{\partial\dot{\sigma}_Z}{\partial\sigma}=0. \end{cases} \tag{3}$$

Equation (3) is the sufficient condition that the components of $\vec{V}_R$, $\vec{V}_M$ and $\vec{V}_Z$ can be represented by the stream functions $R(\lambda,\theta,\sigma)$, $H(\lambda,\theta,\sigma)$ and $W(\lambda,\theta,\sigma)$, respectively, as follows:

$$\begin{cases} u_R=-\dfrac{\partial R}{\partial\theta},\, v_R=\dfrac{1}{\sin\theta}\dfrac{\partial R}{\partial\lambda}, \\[2mm] v_M=-\dfrac{\partial H}{\partial\sigma},\, \dot{\sigma}_M=\dfrac{1}{\sin\theta}\dfrac{\partial(\sin\theta H)}{\partial\theta}, \\[2mm] u_Z=\dfrac{\partial W}{\partial\sigma},\, \dot{\sigma}_Z=-\dfrac{1}{\sin\theta}\dfrac{\partial W}{\partial\lambda}. \end{cases} \tag{4}$$

Because three-pattern circulations (horizontal, meridional and zonal circulations) exist in both the low and middle-high latitudes, the global atmospheric circulation can be expressed as the superposition of the horizontal, meridional and zonal circulations, that is,

$$\vec{V}=\vec{V}_M+\vec{V}_Z+\vec{V}_R, \tag{5}$$

with the following components:

$$\begin{cases} u=u_Z+u_R=\dfrac{\partial W}{\partial\sigma}-\dfrac{\partial R}{\partial\theta}, \\[2mm] v=v_R+v_M=\dfrac{1}{\sin\theta}\dfrac{\partial R}{\partial\lambda}-\dfrac{\partial H}{\partial\sigma}, \\[2mm] \dot{\sigma}=\dot{\sigma}_M+\dot{\sigma}_Z=\dfrac{1}{\sin\theta}\dfrac{\partial(\sin\theta H)}{\partial\theta}-\dfrac{1}{\sin\theta}\dfrac{\partial W}{\partial\lambda}. \end{cases} \tag{6}$$


Equation (5) or (6) is called the three-pattern decomposition model.

In contrast to the traditional two-dimensional decomposition of the atmospheric motion into vortex and divergent parts, the continuity Eq. (5) cannot guarantee the uniqueness of the stream functions $R(\lambda,\theta,\sigma)$, $H(\lambda,\theta,\sigma)$ and $W(\lambda,\theta,\sigma)$ because the three-pattern circulations $\vec{V}_R$, $\vec{V}_M$ and $\vec{V}_Z$ have three spatial dimensions, respectively (Hu et al.,

2017, 2018a, b). The following restriction condition is needed to pick up the correct decomposition (Theorems 1 and 2 in Hu et al., 2018a):

$$\frac{1}{\sin\theta}\frac{\partial H}{\partial\lambda}+\frac{1}{\sin\theta}\frac{\partial(W\sin\theta)}{\partial\theta}+\frac{\partial R}{\partial\sigma}=0. \tag{7}$$

Equation (7) guarantees both the uniqueness of the stream functions $R$, $H$ and $W$ and the physical rationality of the 3P-DGAC method.

Using the 3P-DGAC method, we can rephrase the moisture budget in Eq. (1) to involve the contributions from zonal and meridional circulation. Here, we neglect the relatively smaller terms at low latitudes, including transient eddies, nonlinear effects and surface boundary terms. Thus, we mainly explore the contributions from δTH and δMCD to changes in PmE in this study. Then, the δTH and δMCD can be rewritten as:

$$\delta TH = \underbrace{\underbrace{-\frac{1}{\rho g}\int_0^{p_s}\delta q\nabla\cdot\vec{V}_{R0}\,dp}_{\delta TH_{D\_R}}\underbrace{-\frac{1}{\rho g}\int_0^{p_s}\delta q\nabla\cdot\vec{V}_{Z0}\,dp}_{\delta TH_{D\_Z}}\underbrace{-\frac{1}{\rho g}\int_0^{p_s}\delta q\nabla\cdot\vec{V}_{M0}\,dp}_{\delta TH_{D\_M}}}_{\Delta\delta TH_D}$$

$$\underbrace{\underbrace{-\frac{1}{\rho g}\int_0^{p_s}\vec{V}_{R0}\cdot\nabla\delta q\,dp}_{\delta TH_{A\_R}}\underbrace{-\frac{1}{\rho g}\int_0^{p_s}\vec{V}_{Z0}\cdot\nabla\delta q\,dp}_{\delta TH_{A\_Z}}\underbrace{-\frac{1}{\rho g}\int_0^{p_s}\vec{V}_{M0}\cdot\nabla\delta q\,dp}_{\delta TH_{A\_M}}}_{\delta TH_A} \tag{8}$$

$$\delta MCD = \underbrace{\underbrace{-\frac{1}{\rho g}\int_0^{p_s}q_0\nabla\cdot\delta\vec{V}_{R}\,dp}_{\delta MCD_{D\_R}}\underbrace{-\frac{1}{\rho g}\int_0^{p_s}q_0\nabla\cdot\delta\vec{V}_{Z}\,dp}_{\delta MCD_{D\_Z}}\underbrace{-\frac{1}{\rho g}\int_0^{p_s}q_0\nabla\cdot\delta\vec{V}_{M}\,dp}_{\delta MCD_{D\_M}}}_{\delta MCD_D}$$

$$\underbrace{\underbrace{-\frac{1}{\rho g}\int_0^{p_s}\delta\vec{V}_{R}\cdot\nabla q_0\,dp}_{\delta MCD_{A\_R}}\underbrace{-\frac{1}{\rho g}\int_0^{p_s}\delta\vec{V}_{Z}\cdot\nabla q_0\,dp}_{\delta MCD_{A\_Z}}\underbrace{-\frac{1}{\rho g}\int_0^{p_s}\delta\vec{V}_{M}\cdot\nabla q_0\,dp}_{\delta MCD_{A\_M}}}_{\delta MCD_A} \tag{9}$$


where subscripts D and A represent the terms that are related to divergence and moisture advection, respectively. In addition, the subscripts R, Z, and M indicate the terms that are related to the horizontal, zonal and meridional circulations, respectively. Note that $\vec{V}_R$ represents the horizontal vortex winds, which are nondivergent, which indicates that the terms that are related to the divergence/convergence of $\vec{V}_R$ (i.e., $\delta TH_{D\_R}$ and $\delta MCD_{D\_R}$) are zero. These terms can be clearly seen in Figs.

3(h) and 4(h). In addition, we ignore these two terms in this study.

# 3 Changes in hydroclimate during mid-Pliocene

## 3.1 Changes in precipitation minus evaporation (PmE) in the PlioMIP2 models

The last 100 years of individual PlioMIP2 simulations are used to calculate the multimodel mean (MMM) PmE in Fig. 1 and individual PlioMIP2 models in Fig. 2.

Fig. 1(a) shows that most subtropical regions experience reduced PmE in the mid-Pliocene simulations with respect to the PI simulations, including the subtropical Pacific and subtropical Atlantic in both hemispheres and the subtropical Indian Ocean in the Southern Hemisphere (SH). There is also drying over the South Pacific convergence zone (SPCZ), except in the GISS-E2-1-G, COSMOS and HadGEM3 models (Fig. 2), consistent with other studies evaluating the hydrological cycle response within the PlioMIP2 simulations (Pontes et al., 2020). Note that there is moistening signal in the southern part of the SPCZ in the tropical southern Pacific. In contrast, the increased MMM PmE is located in the deep tropics [i.e., Pacific intertropical convergence zone (ITCZ) and North Indian Ocean], as well as at mid-high latitudes (Fig. 1(a)). However, some models, i.e., the CESM2, GISS-E2-1-G, COSMOS and HadGEM3 models, show a drier Maritime Continent (Fig. 2), which might be related to the changes in Walker circulation (we will discuss this latter in Section 5.3). In addition, the North African and Southeast Asian monsoon regions also show significant moistening signals, which are consistent with faunal remains and palynological transfer functions (Sanyal et al., 2004; Trauth et al., 2007; Xie et al., 2012), as well as with other modeling studies (Zhang et al., 2019a; Li et al., 2020; Feng et al., 2021). Zhang et al. (2016) indicate that the combined influence of SST and $CO_2$ level, as well as the vegetation changes, play a very important role in changing the atmospheric circulation over North Africa during the mid-Pliocene, owing to the increased net atmospheric energy there. Additionally, the expansion of vegetation into the Sahara region tends to decrease the surface albedo, which can enhance the Saharan Heat Low and hence impact rainfall over West Africa, reflecting the vegetation-albedo feedback (Charney, 1975). Recent studies indicate that the enhanced vegetation in PlioMIP2 ensemble is likely to have contributed to increased mid-Pliocene West African summer rainfall (Haywood et al., 2020; Berntell et al., 2021). This change over Southeast Asia is robust among PlioMIP2 models, and only the COSMOS model shows a drier change over East Asia (Fig. 2(e)). Furthermore, the MMM PmE changes over Southeast Asia are mainly focused on the summer time (not shown), suggesting a consequence of strengthened East Asian summer monsoon circulation (Salzmann et al., 2008; Wan et al., 2010; Yan et al., 2012; Zhang et al., 2013; Li et al., 2018; Lu et al., 2021). Note that the mid- to high-latitude North Atlantic becomes drier (Fig. 1(a)).

The response of the hydrological cycle during the mid-Pliocene generally shows a wet-regions-getting-wetter-and-dry-regions-getting-drier pattern, especially over the ocean. These features are apparent in the zonal average of the PmE change (Fig. 1(b)), except in the GISS-E2-1-G model (Fig. 2(f)). The tropical regions become wetter, and subtropical regions become drier, which are similar to the results from future high-$CO_2$ scenario experiments (Chou et al., 2009). Earlier studies indicate these features of changes in PmE at low latitudes are linked to the increased specific humidity (i.e., changes in thermodynamic effect). However, there are some opposite phenomenon as well, when looking the regional changes in PmE

(i.e., north Indian Ocean, North Africa and SPCZ). These may suggest that another factor, such as atmospheric circulation anomalies (i.e., changes in dynamic effect), may play an important role in changing regional PmE pattern at low latitudes.

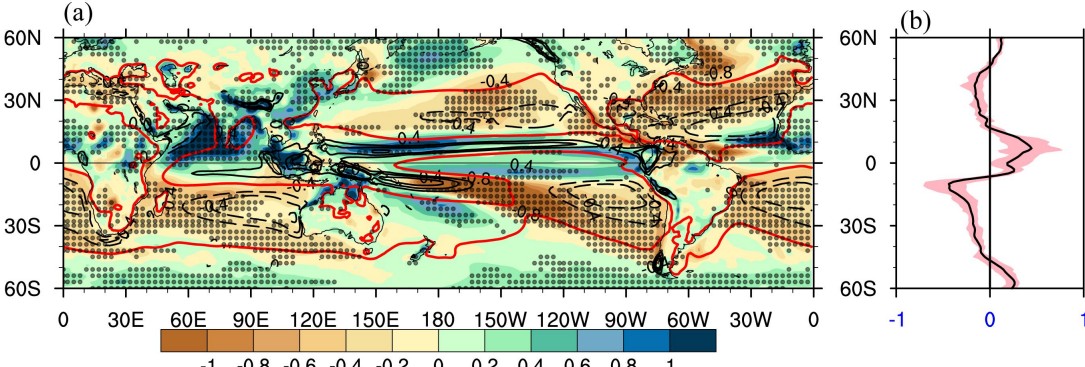

**Figure 1: (a) Changes in multimodel mean (MMM) PmE for the mid-Pliocene compared with the PI simulation (shading), overlaid by the climatological MMM PmE of the PI simulation (for the contours, a solid line indicates positive values and a dashed line indicates negative values). The red solid curves represent the zero value. (b) The zonal average of the change in PmE, where the shading indicates the interquatile range among models. Stippling (left) indicates regions where at least 10 of 13 simulations in the model group agree on the sign of the ensemble mean. Units: mm·day$^{-1}$.**

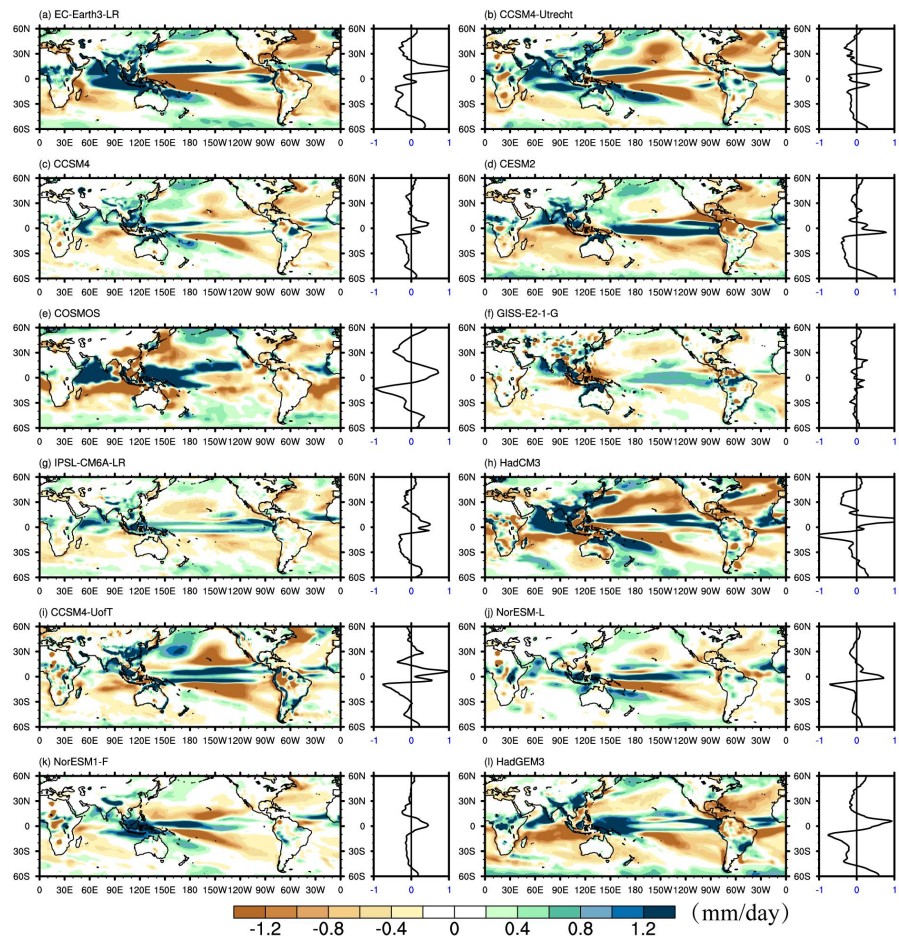

**Figure 2: The simulated PmE changes in individual PlioMIP2 models. The zonal average of the PmE changes in each model is shown on the right. Units: mm·day⁻¹.**

### 3.2 Previous model-data comparisons of hydrological changes in the PlioMIP2 ensemble

Multi-proxy studies are qualitatively consistent with the results of PlioMIP2 ensemble (Feng et al., 2021). For several studies proxy reconstructions suggest an expansion of woodland and a higher density of land cover over northern Africa, indicating moistening signals there (Salzmann et al., 2008; Bonnefille, 2010). The sedimentological indicators and pollen data also suggest a more humid climate over the Levant and Arabian Peninsula during the mid-Pliocene (Munoz et al., 2002; Heermance et al., 2013). In addition, the faunal remains or palynological transfer functions show wetter conditions in East

and South Asia during the mid-Pliocene (Sanyal et al., 2004; Igarashi and Yoshida, 1988; Kou et al., 2006). However, there still remains uncertainties related to hydroclimate of the mid-Pliocene. For instance, pollen evidence suggest little hydroclimate change during the Pliocene in Qaidam Basin and southwest China's Yanmou region (Wang et al., 1999; Chang et al., 2010; Heermance et al., 2013). Some proxies even show a drier climate over the Loess Plateau region (Ji et al., 2017;

Sun et al., 2010). Note that the relatively low availability of Pliocene hydroclimate proxies make it difficult to perform a model-proxy comparison. Besides, PlioMIP2 modeling experiments are designed to simulate the Marine Isotope Stage KM5c (3.205 Ma) during mid-Pliocene, and this particular orbital interval might can not represent the full Pliocene hydroclimate variability, adding uncertainties to model-proxy comparison (Samakinwa et al., 2020).

## 4 Thermodynamic and dynamic contributions to changes in PmE

Moisture budget analyses are conducted to shed light on the mechanisms driving the changes in PmE during the mid-Pliocene. Based on this decomposition, the changes in PmE are mainly influenced by the changes in humidity with unaltered atmospheric circulation (called the thermodynamic term, δTH) and change in atmospheric circulation with no change in humidity (called the dynamic term, δMCD) at low latitudes. The thermodynamic term (δTH) and its decomposition are plotted in Fig. 3. It is clear that δTH captures the main features of hydrological cycle change (Figs. 3(a) vs 1(a)). That is, the positive and negative contributions over the already convergent (i.e., ITCZ and SPCZ) and divergent (subsidence of local Hadley circulation) regions, respectively. In general, the thermodynamic term increases PmE by ~58.6% over tropic, and decreases PmE by ~84.6% over subtropics (not shown), respectively. This term does not alter the spatial distribution of climatological PmE (contours in Fig. 1(a)) but amplifies the intensity of the existing pattern of PmE, reflecting the wet-getting-wetter-and-dry-getting-drier mechanisms (Held and Soden, 2006). The results are consistent with future global warming scenarios (Chou et al., 2009; Wang et al., 2012; Li et al., 2013).

From the perspective of global atmospheric circulation, previous studies have indicated that global atmospheric circulation can be decomposed into a superposition of horizontal, meridional, and zonal circulations (Hu et al., 2017, 2018a, b). δTH is further decomposed by using the 3P-DGAC method (Fig. 3(c)-(k)). The estimated δTH in Fig. 3(b), calculated by the sum of the right-hand side in Eq. (8) of the 3P-DGAC decomposition method shows a similar distribution to the δTH field shown in Fig. 3(a) with a pattern correlation coefficient (PCC) of 0.80. This result indicates that the decomposition is representative. At low latitudes, the δTH mainly comes from terms that are related to climate mean meridional and zonal circulation (Fig. 3(c) and (d)); while at mid-high latitudes, the δTH mainly comes from horizontal circulation (Fig. 3(e)). It is clear that the thermodynamic changes associated with meridional circulation can explain the large portion of δTH (PCC of 0.9) at low latitudes, which is caused by increased specific humidity within the divergence of climate mean meridional circulation ( $\delta TH_{D\_M}$ ; Fig. 3(f)). The zonal circulation can also explain δTH to some extent, with a positive contribution mainly over Maritime Continent extending eastward to the equatorial central Pacific and eastern coast of North/South America, and a negative contribution over the eastern Pacific extending from the western Indian Ocean to the Greater Horn of Africa and the eastern tropical Atlantic (Fig. 3(d)). These changes associated with zonal circulation are linked to the increased specific humidity with divergence of the mean zonal circulation ( $\delta TH_{D\_Z}$ ; Fig. 3(g)). At mid-high latitudes, the δTH induced by climate mean horizontal circulation is caused by changes in moisture advection (Fig. 3(k)), e.g., Western

coast of North America, a region extending from the Southern tip of South America to the Central tropical Pacific Ocean, and Southern tip of South Africa.

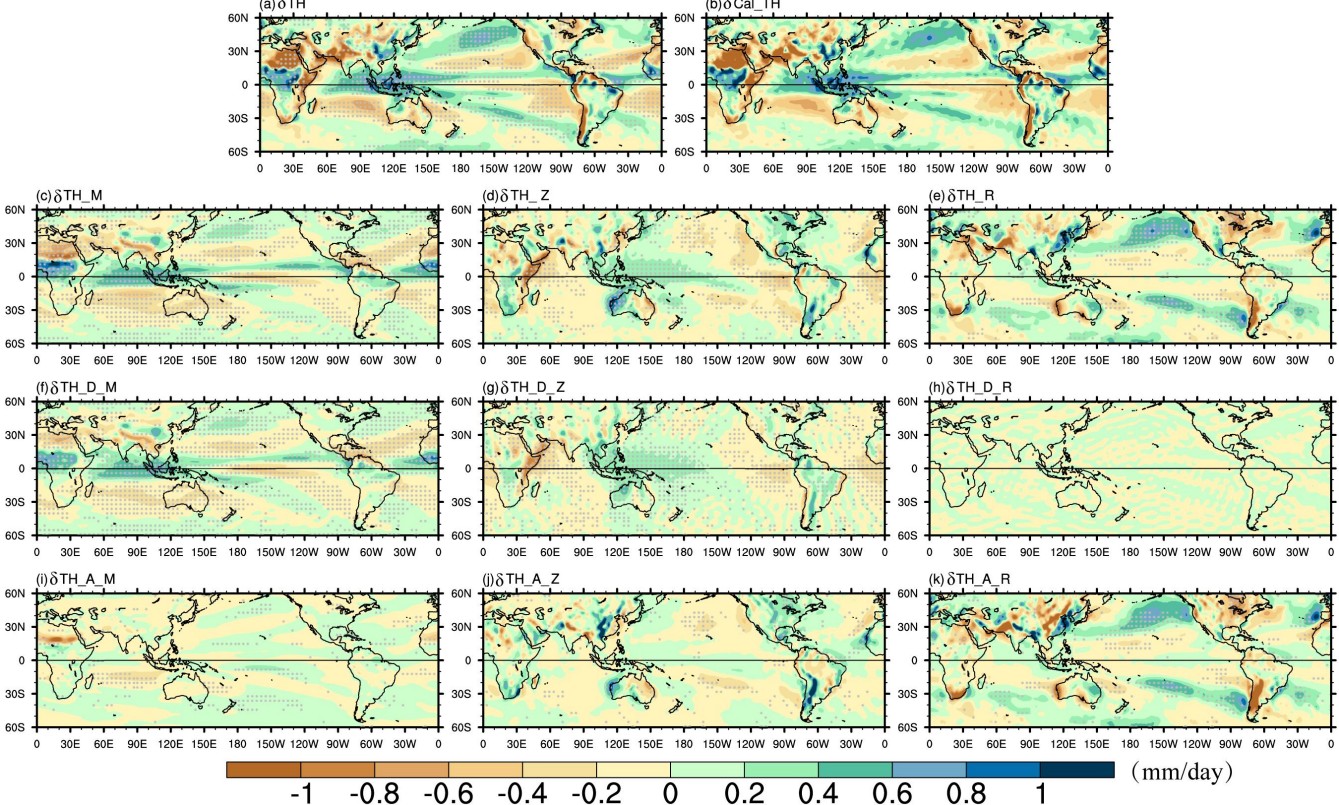

**Figure 3: The annual mean changes in the moisture budget components of the mid-Pliocene minus PI control of the PlioMIP2 multimodel mean, which reflect (a) the thermodynamic term and (b) the estimated change in the thermodynamic term [calculated by the sum of the right-hand side terms in Eq. (8) of the 3P-DGAC decomposition method]. The contributions to the change in the thermodynamic: the thermodynamic term induced by the climate mean (c) meridional, (d) zonal and (e) horizontal circulations. The corresponding changes in (f-h) and (i-k) are the terms in (c-d) the component related to divergent mean flow and change in moisture advection, respectively. Stippling indicates regions where at least 10 of 13 simulations in the model group agree on the sign of the ensemble mean. Units: mm·day$^{-1}$.**

It is evident that the δTH component does not describe the full contribution to the changes in PmE, especially over the North African and Southeast Asian monsoon regions, SPCZ and North Indian Ocean, where we must consider the dynamic effect. The dynamic effect (δMCD), reflecting the impact of circulation changes, partially offsets the δTH at low latitudes (Fig. 4(a)). In particular, δMCD reduces PmE in the deep tropics, i.e., the ITCZ, SPCZ and Maritime Continent. In

contrast, δMCD can moisten subtropical regions, especially over the subtropical eastern Pacific, southern Indian Ocean and Atlantic Ocean of both hemispheres. Compared with δTH, the dominating contribution from δMCD to changes in PmE lies adjacent to the North Indian Ocean, SPCZ and the North African and Southeast Asian monsoon regions (Fig. 4(a)).

The estimated δMCD in Fig. 4(b), calculated by the sum of the right-hand side terms in Eq. (9) of the 3P-DGAC decomposition method, is consistent with the δMCD in Fig. 4(a) with a PCC of 0.93. This result indicates that the

decomposition is representative. The anomalous divergence of the meridional circulation component ( $\delta MCD_{D\_M}$ ) appears to dry the deep tropics but moisten its Northern hemispheric part of the deep tropics, which is associated with the northward shift of the ITCZ (Fig. 7(c)). In particular, the northward shift of the ITCZ is clear from 150°E to the east in the Pacific (Fig. 4(f)). In addition, the component contributes a large portion to enhance PmE over the North African and Southeast Asian monsoon regions. However, the tropical southern Pacific is even more complicated. $\delta MCD_{D\_M}$ term contributes to

reduced PmE over SPCZ region, but increased PmE over the southern part of SPCZ in the tropical southern Pacific. A previous study suggested that these changes in PmE followed the southward shift of the SPCZ, which was mainly modulated by the intensified and westward shift of the South Pacific subtropical high for the mid-Pliocene compared with the PI simulation (Pontes et al., 2020). For the adjacent north Indian Ocean, the convergence of zonal circulation anomalies ( $\delta MCD_{D\_Z}$ ) is the first-order contribution to strengthen the dynamic effect (by ~45%) and hence enhances the PmE (Fig.

4(g)).

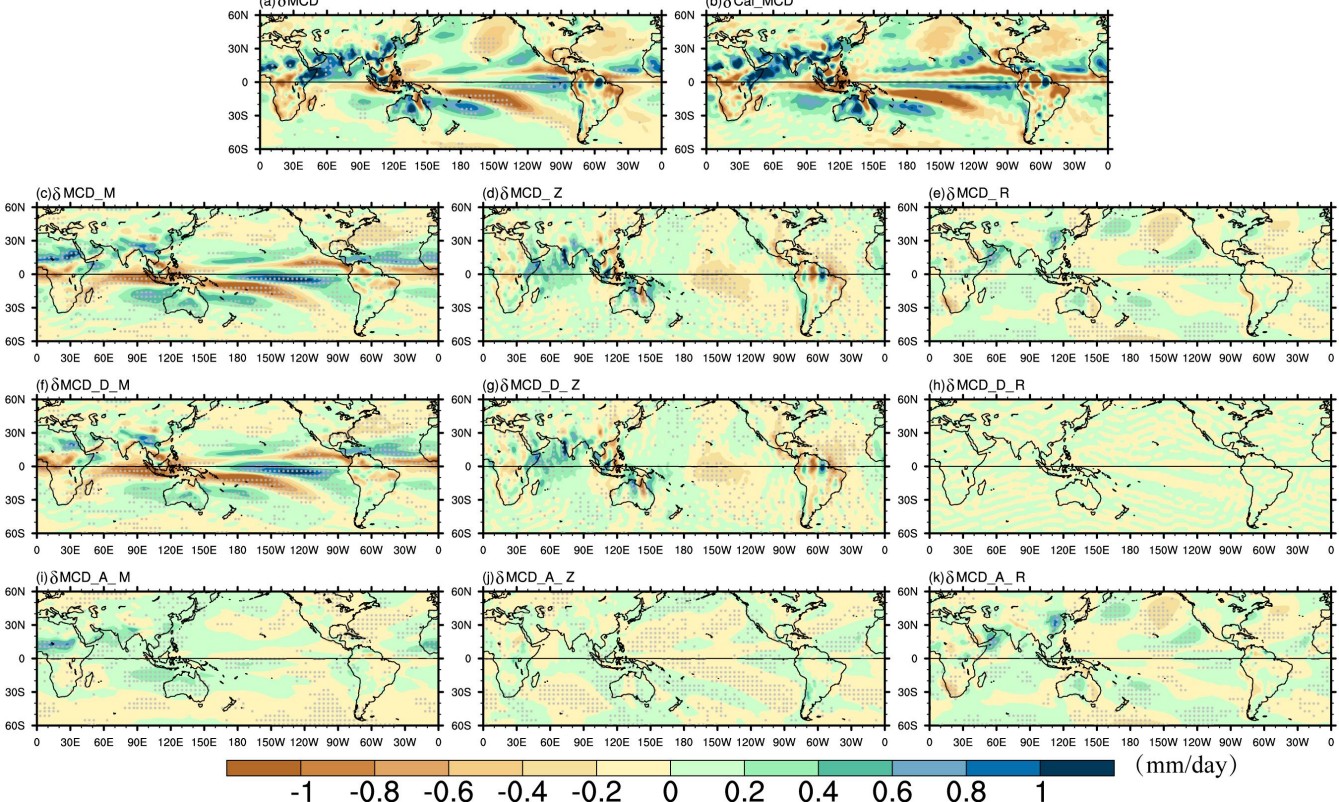

**Figure 4: The annual mean changes in the moisture budget components of the mid-Pliocene minus PI control of the PlioMIP2 multimodel mean, which reflect (a) the dynamic term and (b) the estimated change in dynamic term [calculated by the sum of the**

**right-hand side terms in Eq. (9) of the 3P-DGAC decomposition method]. The contributions to the dynamic: the dynamic term induced by the anomalous (c) meridional, (c) zonal, and (d) horizontal circulations. The corresponding changes in (f-h) and (i-k) are the terms in (c-d) the component related to divergent mean flow and change in moisture advection, respectively. Stippling indicates regions where at least 10 of 13 simulations in the model group agree on the sign of the ensemble mean. Units: mm·day⁻¹.**

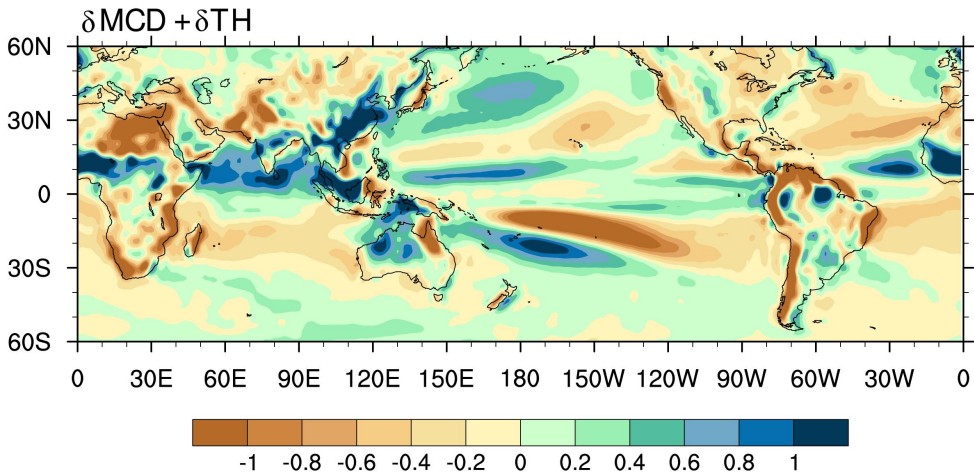

**Figure 5: The estimated annual mean changes in the PmE (calculated by the sum of the dynamic term and thermodynamic term) of the mid-Pliocene minus PI control of the PlioMIP2 multimodel mean. Unit: mm day⁻¹.**

In summary, the dynamic and thermodynamic terms can explain the largest changes in PmE at low latitudes (Figs. 5 vs 1). The thermodynamic term induced by the divergence of the mean meridional circulation is the dominant process driving changes in PmE at low latitudes (Fig. 3(f)). However, the dynamic term partially offsets δTH, especially over the ITCZ, SPCZ and Maritime Continent, via changes in the divergence of meridional circulation. Even the dynamic term overwhelmingly contributes to the increased PmE over North African and Southeast Asian monsoon regions and the North Indian Ocean (Figs. 4 vs 5). Note that the former two are mainly caused by meridional circulation anomalies, but the latter is dominated by zonal circulation anomalies.

We further decompose the meridional moisture transport into terms that reflect the changes in specific humidity (meridional moisture transport induced by the thermodynamic effect; MMTT) and circulation (meridional moisture transport induced by the dynamic effect; MMTD) in Fig. 6(a). As expected, all models show that the MMTT is responsible for the wetter tropics and drier subtropics in the mid-Pliocene simulation, indicating a dry-gets-drier-and-wet-gets-wetter mechanism. These features are robust among models and are associated with the increased specific humidity combined with the mean meridional circulation from the PI control (Fig. 6(b)), as mentioned above. This is because the zonal-mean wind depicts southerly (northerly) wind between the equator and subtropical SH (Northern Hemisphere; NH) for the climate mean meridional circulation in the PI simulations. When the climatological wind is combined with increased specific humidity in the low-level troposphere (Fig. 7(b)), more moisture is transported from the subtropics to the tropics, resulting in drier subtropics and wetter tropics. In contrast, the MMTD shows a large spread across PlioMIP2 models. On average, the anomalous MMTD appears to weaken thermodynamic contributions in the subtropical NH but strengthen it in the

subtropical SH via meridional circulation anomalies (Fig. 6(c)). This indicates that the changes in MMTD favor the transport
of more (less) moisture from the tropics to the NH (SH) subtropics, which is caused by the northward shift of the meridional
circulation (as detailed further in Section 5.2). The equatorward moisture transport anomalies of SH's dynamic component
are due to anomalous southerly winds in the subtropical southern Pacific (Fig. 9). This feature acts to dry the SPCZ and
moisten the south SPCZ and the equatorial central-eastern Pacific (Fig. 4(f)).

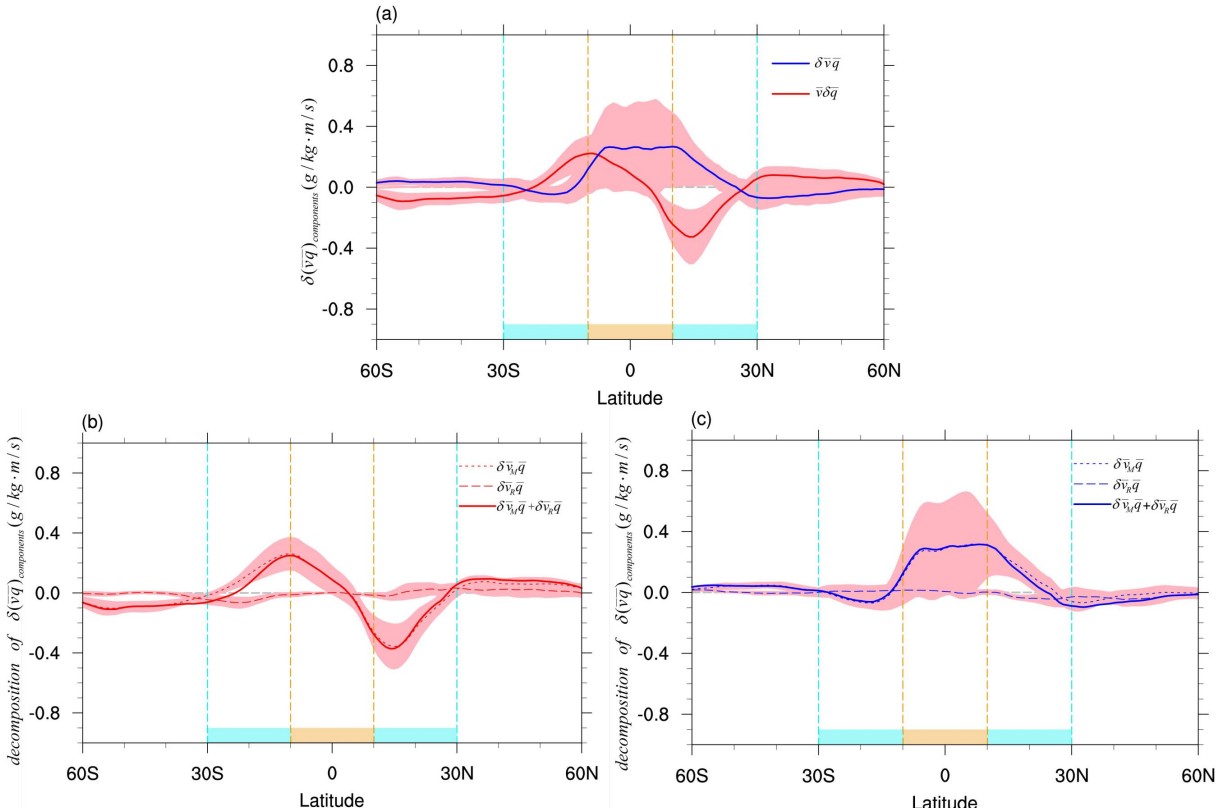

**Figure 6: (a) The time-mean meridional moisture transport anomalies induced by thermodynamic and dynamic effects for the
mid-Pliocene compared with PI simulations. (b) and (c) are thermodynamic and dynamic terms in (a) that are induced by the
meridional wind $\vec{V}_M$ and $\vec{V}_R$ decomposed from the 3P-DGAC method, respectively. Here, the tropical region is defined as the
region between 10°S and 10°N (marked as an orange band), while the subtropical region refers to 10-30° N and 10-30° S (marked**
**as an cyan band). The shading indicates 1 standard deviation of individual models departing from the MMM. Units: g•kg⁻¹•m•s⁻¹.**

## 5 Mechanisms for the changes in moisture budget components

Thus far, we have shown that the anomalous hydroclimate within the mid-Pliocene simulations involves anomalies of both
thermodynamic and dynamic effects at low latitudes. In this section, we further examine the corresponding mechanisms in
turn.

## 5.1 Changes in specific humidity

Fig. 7(a) shows the changes in MMM SST superimposed on the reconstructed SST anomalies (McClymont et al., 2020). In the MMM, SSTs range from between 1 and 6°C warmer in the mid-Pliocene simulations than in the PI simulations. Note that the SST warming is amplified in the northwest tropical Indian Ocean, whereas it is reduced off the Indonesian coast, showing tropical Indian Ocean dipole (IOD)-like pattern. The sharp SST gradients drive strong southeasterly wind anomalies on the equator (not shown). Xie et al. (2010) suggest that this easterly wind anomaly may shoal the thermocline in the east, helping lower SST there via upwelling, and indicating this SST anomaly over tropical Indian Ocean may be related to the Bjerknes feedback. The simulated North Atlantic warming might be related to an intensified mid-Pliocene AMOC (Li et al., 2020). However, Zhang et al. (2021) suggest that the increased background ocean vertical mixing parameters could also contribute to the warm SSTs there. In addition, the relative smaller SST warming in Southeast Pacific and Atlantic, which is collocated with the intensified southeast trades, suggests the role of wind-evaporation-SST feedback (Xie et al., 2010). These SST warming patterns are consistent with current studies (Haywood et al., 2020; Williams et al., 2021). As expected, the specific humidity is increased in the low-level troposphere in the mid-Pliocene warm period (Fig. 7(b)) (Murray, 1966; Held and Soden, 2006). On the other hand, the sinking branch of meridional circulation in the control climate is located in subtropical regions, showing divergent circulation $\nabla \cdot \vec{V} > 0$ in the low-level troposphere. And the contrary applies for the regions of deep tropics, i.e., ITCZ and SPCZ. These two factors contribute the $\delta TH_{D\_M}$ term (i.e.,

$-\frac{1}{\rho g} \int_0^{p_s} \delta q \nabla \cdot \vec{V}_{M0} dp$ ) to the thermodynamic effect (Figure 3f) and hence changes in PmE.

Although the $\delta TH_{D\_M}$ term is the first-order to control the thermodynamic effect in most regions, $\delta TH_{D\_Z}$ term contributes to the thermodynamic effect to some extent, especially over the adjacent north Indian Ocean. The climate mean zonal circulation characterizes ascending in the tropical western Pacific, tropical African and tropical southern American regions, favoring convergent circulation (i.e., $-\nabla \cdot \vec{V}_{Z_0} > 0$ ) there (Fig. 6(d); Hastenrath, 1991; Peixoto and Oort, 1992).

With increased specific humidity ( $\delta q > 0$ ) under a warmer climate, the $\delta TH_{D\_Z}$ term (i.e., $-\frac{1}{\rho g} \int_0^{p_s} \delta q \nabla \cdot \vec{V}_{Z0} dp$ ) shows positive contribution and hence increase PmE in these regions (Figure 3g). In the contrary, the $\delta TH_{D\_Z}$ favors to decrease PmE over the western Indian Ocean, eastern Pacific and tropical Atlantic (Figure 3g), where the climate mean zonal circulation is divergent (Figure 7d).

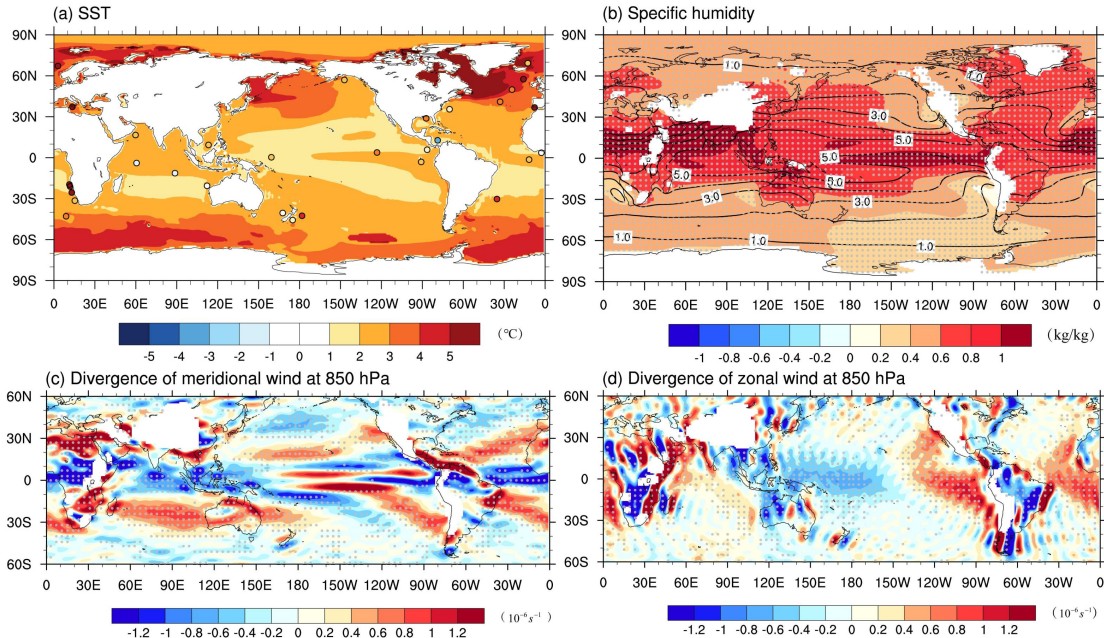

**Figure 7: Change in (a) MMM SST (shading, unit: ℃), (b) specific humidity (shading, unit: kg•kg⁻¹) overlaid by its climate mean for PI simulation. (c) and (d) show the MMM divergence of meridional $\vec{V}_M$ and zonal wind $\vec{V}_Z$ fields decomposed from the 3P-DGAC method at the 850 hPa level for PI simulation (unit: 10⁻⁶ s⁻¹). The circles in (a) are the anomalies of reconstructed SST (McClymont et al., 2020) from the alkenone-derived $U_{37}^{K'}$ index (Prahl and Wakeham, 1987) and foraminifera calcite Mg/Ca (Delaney et al., 1985). Stippling in (b-d) indicates regions where at least 10 of 13 simulations in the model group agree on the sign of the ensemble mean.**

### 5.2 Response in meridional circulation

In Section 4, we have demonstrated that the primary dynamic contribution to changes in PmE is a consequence of anomalous meridional circulation ( $\delta MCD_{D\_M}$ term). Fig. 8(a) shows the annual mean mass stream function (MSF) of meridional

circulation for PI simulation (contours), which is similar to present-day meridional circulation (Cheng et al., 2020). During the mid-Pliocene, the meridional circulation changes are characterized by enhanced meridional circulation in the SH tropics and weakened meridional circulation in the NH tropics (shading in Fig. 8), which is caused by the northward shift of meridional circulation in the SH, as indicated in our later discussion. To quantify meridional circulation changes, we further calculated the intensity in Fig. 8(b). The intensity is defined as the maximum of the absolute average MSF between 200 hPa

and 925 hPa in the range of 30°S to 30°N (Oort and Yienger, 1996) in Fig. 8(a). Models simulate a consistently weakened meridional circulation intensity in the NH and a slightly strengthened intensity in the SH (Fig. 8(b)), which is related to the hemispheric asymmetry of the atmospheric energy budget (Feng et al., 2020).

As a result, meridional circulation anomalies could induce divergent/convergent circulation anomalies in the low-level troposphere (Fig. 9). The weakened local meridional circulation leads to anomalous southerly winds spanning

northeastern South America eastward to the northwestern Pacific region. These meridional circulation anomalies induce the anomalous divergence (convergence) of circulation over the Indo-Pacific warm pool (adjacent to subtropic regions), resulting in a negative (positive) contribution from $\delta MCD_{D\_M}$ (Figs. 9 vs 4(f)). In fact, this anomalous meridional circulation is closely related to the strengthened Asian summer monsoon (not shown), consistent with previous studies (Zhang et al., 2013; Prescott et al., 2019). In addition, anomalous northerly winds exist in the western tropical Pacific, but southerly winds are located in the central Pacific (Fig. 9). These circulation anomalies could induce $\delta MCD_{D\_M}$, which favors moistening of the equatorial central Pacific and southern part of the SPCZ region but dries the SPCZ (Fig. 4(f)). Previous studies have indicated that these circulation anomalies are caused by the southward shift of the SPCZ, which is mainly modulated by the intensified and westward shift of the South Pacific subtropical high for the mid-Pliocene compared with the PI simulation (Pontes et al., 2020).

One question arises as to what causes the meridional circulation changes in mid-Pliocene conditions. At low latitudes, it is worth noting that the ITCZ lies at the foot of the ascending branch of the meridional circulation, which is highly linked to the hemispheric asymmetry of the atmospheric energy budget (Frierson et al., 2013). We further quantify the shift of the ITCZ in Fig. 8(c) and Earth's energy budget in Fig. 8(d) and (e). The definition of the ITCZ location is the latitude of the maximal annual mean precipitation between 20°S and 20°S (Frierson and Hwang, 2012; Donohoe et al., 2013). On average, ensemble models show that the NH atmosphere receives 1.5 W · m$^{-2}$ more net radiation than the SH (Fig. 8(d)), which could induce an increased cross-equatorial southward energy flux of 0.22 PW (Fig. 8(e)). Thus, this imbalance in the atmospheric energy budget causes a 1.1° northward shift in the zonal-mean ITCZ latitude. Consequently, this shift of the ITCZ reorganizes atmospheric circulation (Watt-Meyer and Frierson, 2019), leading to the northward movement of the meridional circulation in the SH (Fig. 8(a)). This meridional circulation shift could result in a weakened (strengthened) meridional circulation in the NH (SH) (Fig. 8(b)) and hence drive $\delta MCD_{D\_M}$ (Fig. 4(f)) and MMTD (Fig. 6(c)). Pontes et al. (2021) indicate that the anomalous wind over southern Pacific is related to the ENSO weakening across models, which could favor to northward shift of ITCZ. In addition, it should be noted that the northward shift of ITCZ exist in both boreal summer (JJA) and winter (DJF) season, accompanied with the northward shift of meridional circulation in the SH (not shown). Pontes et al. (2021) further suggest that the northward shift of Pacific ITCZ during austral spring-summer is remarkably related to the ENSO weakening across models, which is associated with the stronger climatological circulation in the SH.

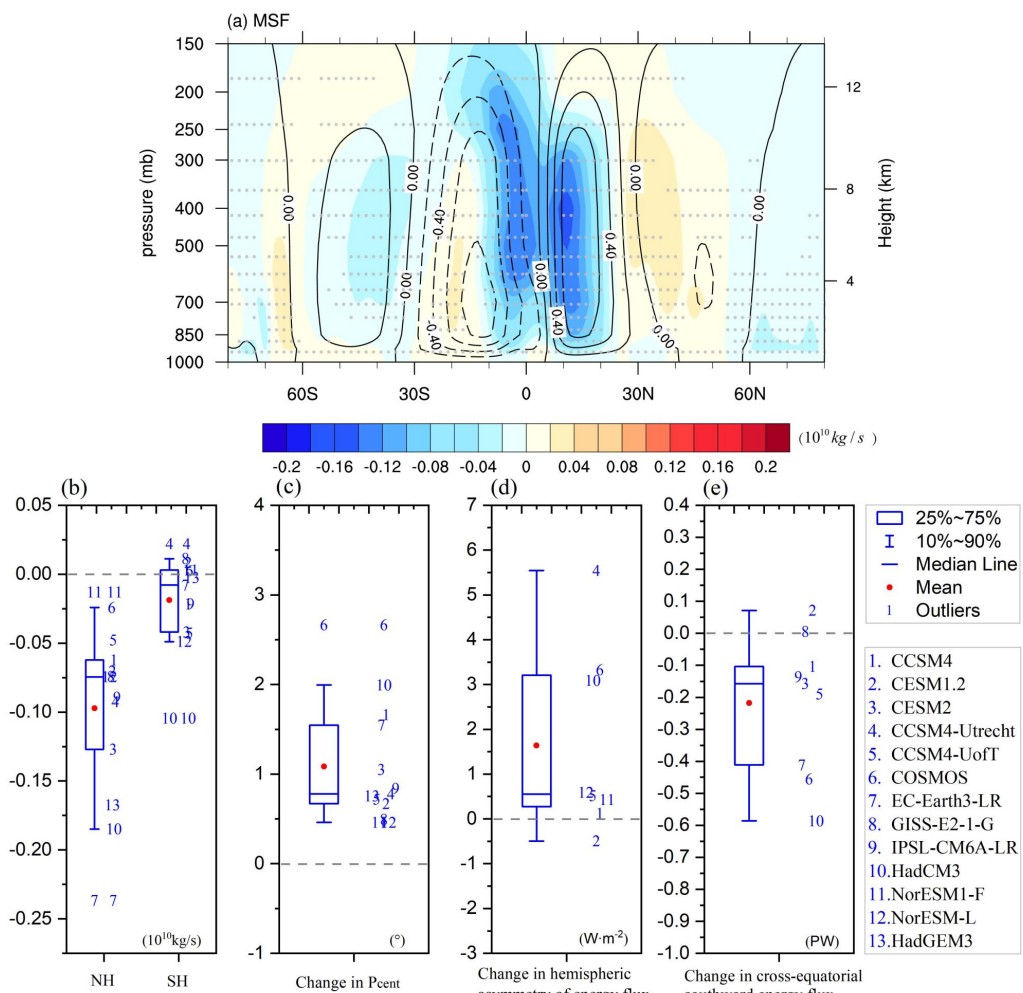

**Figure 8: (a) Changes in annual mean MSF (shading; units: $10^{10}$ kg•s$^{-1}$) of meridional circulation for mid-Pliocene with respect to the PI simulation, overlaid by the climate mean MSF for the PI simulation (contours). The meridional wind $\vec{V}_M$ is decomposed from the 3P-DGAC method. Solid curves indicate positive values, and dashed curves indicate negative values. Stippling indicates regions where at least 10 of 13 simulations in the model group agree on the sign of the ensemble mean. (b) Changes in annual mean intensities (unit: $10^{10}$ kg/s) of meridional circulation in the NH and SH. (c) The latitudes of the center of annual mean precipitation between 20°S and 20°N (unit: °). (d) Hemispheric asymmetry (NH minus SH) of energy flux into the atmosphere (unit: W•m$^{-2}$). (e) Changes in the integrated atmospheric meridional heat transport across the equator (unit: PW).**

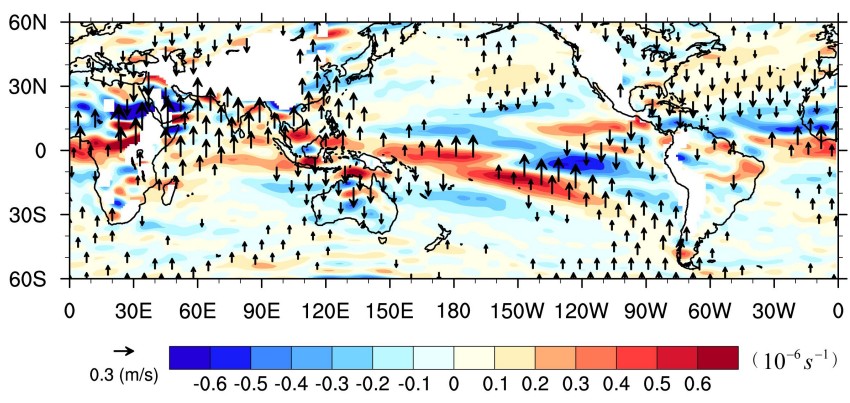


**Figure 9: The changes in meridional wind $\vec{V}_M$ decomposed from the 3P-DGAC method at the 850 hPa level (vectors, units: m•s⁻¹), overlaid by its divergent circulations (shading, units: 10⁻⁶ s⁻¹). Only vectors where at least 10 of 13 simulations in the model group agree on the sign of the ensemble mean are shown here.**

**5.3 Response in zonal circulation**

As mentioned above, $\delta MCD_{D\_Z}$ plays a key role in the changes in PmE over the northern Indian Ocean. As this term is

linked to Walker circulation anomalies, we further discuss Walker circulation changes in the mid-Pliocene warm climate.

There is a noticeable diversity in the simulated Pacific Walker circulation (PWC) intensity across the models (Fig. 10(a)). In addition, previous work has suggested that the PWC intensity is closely tied to the zonal SST and SLP gradient during the mid-Piacenzian (Tierney et al., 2019). In this paper, the dSLP and dSST are defined as the difference in SLP and

SST across the equatorial Indo-Pacific (160°W-80°W, 5°S-5°N minus 80°E-160°E, 5°S-5°N). As expected, the models with an enhanced zonal SST gradient across the equatorial Indo-Pacific tend to produce weaker zonal SLP gradient and decreased PWC (not shown), with the inter-model correlations of -0.95 and -0.75, respectively. Note that the PlioMIP2 models produce a large spread in simulating the changes in dSST (Fig. 10(c)) and dSLP (Fig. 10(d)), which is consistent with the results in Fig. 10(a). Previous studies have suggested that the east-west SST gradient was reduced in SST proxies (Tierney et al., 2019).

This feature is captured by the CESM2, GISS-E2-1-G and HadGEM3 models (Fig. 10(a)-(c)). However, other models, i.e., the CCSM4, CCSM4-Utrecht, EC-Earth3-LR, and NorESM-L models, consistently simulated stronger PWC intensity (Fig. 10(a)-(c)). That is, the results suggest that the model-simulated changes in the strength of PWC are probably highly model dependent, which might be affected by the different parameterizations (Tierney et al., 2019).

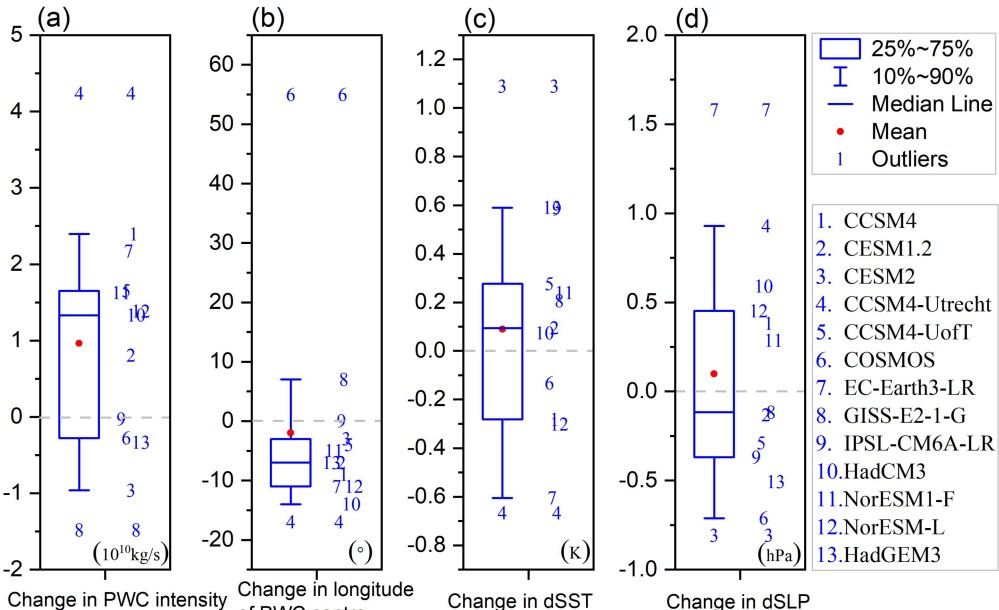

**Figure 10: Changes in annual mean of the (a) intensities of Pacific Walker circulation (PWC; unit: $10^{10}$ kg/s), (b) longitude of the PWC cell center (units: °), (c) dSST (units: K), and (d) dSLP (units: hPa) for mid-Pliocene simulations compared with the PI simulation. Here, the PWC intensity is defined as the vertically integrated ZMS (Bayr et al., 2014; Schwendike et al., 2014) averaged in the equatorial Pacific (140°E-120°W), and the location of the PWC cell center is the longitude of the maximum ZMS. The dSLP and dSST are defined as the difference in SLP and SST across the equatorial Indo-Pacific (160°W-80°W, 5°S-5°N minus**
**80°E-160°E, 5°S-5°N).**

However, the westward shift of PWC is a robust feature among these models except the COSMOS and GISS-E2-1-G models (Fig. 10(b)). To discuss the impact of the PWC shift on atmospheric circulation in the tropics, we further calculate the changes in the zonal mass stream function (ZMS) for the mid-Pliocene with respect to the PI simulation in Fig. 11. As

suggested in Fig. 7(d), the ZMS in the PI simulation (contours in Fig. 11(a)) is characterized by ascending in the tropical western Pacific and Maritime Continent and descending in the western Indian Ocean and eastern Pacific, consistent with previous studies (Kamae et al., 2011; Bayr et al., 2014; Ma and Zhou, 2016; Han et al., 2020). Compared with the PI simulation, the most striking features in the mid-Pliocene simulation are weakened ascending over the Maritime Continent and tropical western Pacific and strengthened descending on the western Indian Ocean, indicating a westward expansion of

the PWC (Fig. 11(b)).

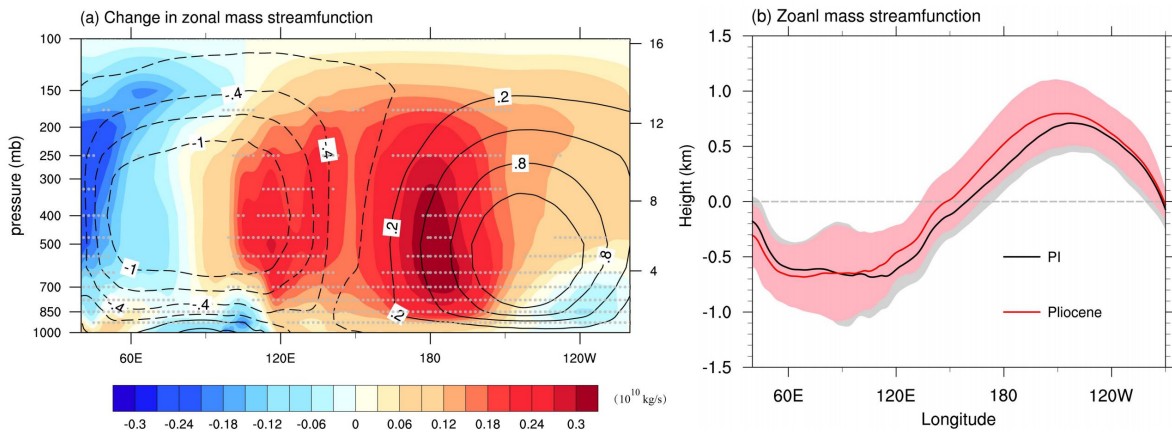

**Figure 11: (a) Changes in ZMS (shading, unit: 10<sup>10</sup> kg s<sup>-1</sup>) averaged between 10°S/N for the mid-Pliocene with respect to the PI** simulation, overlaid by the climate mean ZMS for the PI simulation (contours). The zonal wind $\vec{V}_Z$ is used to calculate ZMS, which is decomposed from the 3P-DGAC method. The contours represent the climate mean ZMS for the PI simulation. Solid curves indicate a positive value, and dashed curves show a negative value. Stippling indicates regions where at least 10 of 13 simulations in the model group agree on the sign of the ensemble mean. (b) is the vertical integrated ZMS in (a). The gray and pink shading indicates 1 standard deviation of individual models departure from the MMM mean of MSF for the PI and mid-Pliocene simulations, respectively.

The westward shift of the PWC can also be seen from the potential velocity (Fig. 12). This shows that the center of anomalous positive values is located in the northern Indian Ocean. In contrast, the center of a negative value exists in the equatorial eastern Pacific and western Atlantic in the low-level troposphere (Fig. 12(a)). Concurrent, generally opposite anomalies can be seen in the upper-level troposphere (Fig. 12(b)). Indeed, these features indicate an upward (downward) motion shift from the tropical western Pacific (eastern Pacific) to the west of the Indian Ocean (central Pacific), resulting

from the westward expansion of the PWC (Figs. 10(b) and 11). That is, when divergent/convergent circulations are combined with the climate mean specific humidity ( $\overline{q} > 0$ ) in the lower troposphere, they can trigger a negative/positive contribution from the $\delta MCD_{D\_Z}$ term to changes in PmE (Fig. 4(g)).

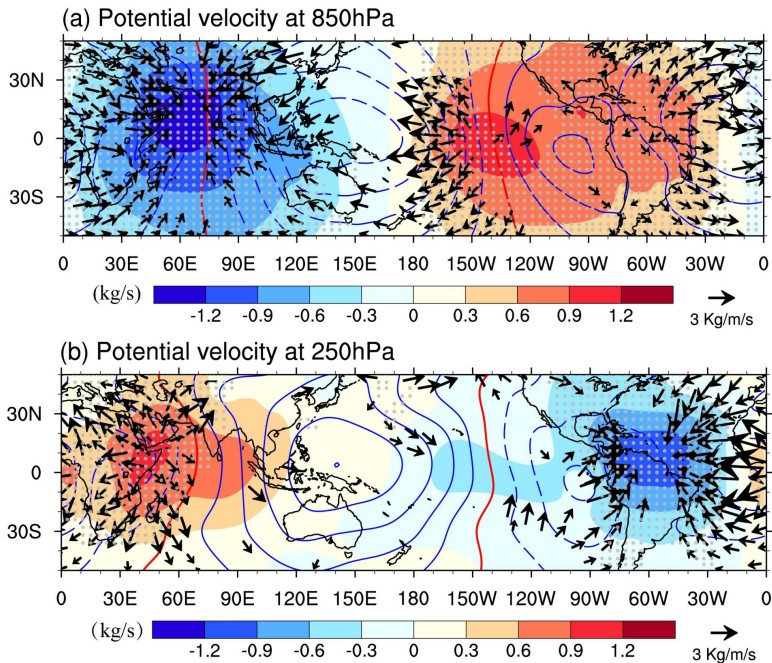

**Figure 12: Changes in the potential function of zonal wind $\vec{V}_Z$ at (a) 850 hPa and (b) 250 hPa (shading, unit: kg/s), corresponding to the divergent mode of the wind field (vectors, unit: kg/m/s). $\vec{V}_Z$ is decomposed from the 3P-DGAC method. The contours represent the climate mean of the potential function for the PI simulation. Solid curves indicate positive values, and dashed curves indicate negative values. Red solid curves represent zero values. The vectors and stippled regions are where at least 10 of 13 simulations in the model group agree on the ensemble mean.**

## 6 Discussion and conclusions

This paper evaluates the changes in the large-scale hydrological cycle during the mid-Pliocene with respect to the PI based on 13 PlioMIP2 simulations. A diagnostic analysis using the moisture budget equation and the Earth's energy budget provides insight into the mechanisms. The main conclusions are summarized as follows.

The PlioMIP2 models show large spatial differences in PmE. The MMM generally depicts a wet-regions-getting-wetter (i.e., ITCZ, Maritime Continent and monsoon regions) and-dry-regions-getting-drier (i.e., a sinking branch Hadley circulation) pattern during the mid-Pliocene warm climate. According to the moisture budget equation, a large part of the changes in PmE at low latitudes are due to the increased specific humidity. However, the thermodynamic component cannot fully explain the changes in PmE. The dynamic effects offset the thermodynamic effects to some extent and even determine a larger contribution to the changes in PmE in the southern tropical Pacific and northern Indian Ocean. We find increased hemispheric asymmetries of the atmospheric energy budget (larger atmospheric energy over NH than SH) during the mid-Pliocene compared with PI, which could induce the northward shift of the ITCZ and reorganize atmospheric circulation. These features can result in a weakening meridional circulation in the Northern Hemispheric monsoon regions and a

strengthening meridional circulation in the SH. In addition, the anomalous meridional circulation can dry the deep tropics but moisten the northern part of the ITCZ. Furthermore, these anomalies dry the SPCZ region and wet its southern part, which is associated with the southward shift of the SPCZ. We also find a robust westward shift in PWC, which appears to moisten the northern Indian Ocean via anomalous convergence of zonal circulation.

   Our analyses provide a relatively complete understanding of the changes in the large-scale hydrological cycle within the PlioMIP2 ensemble. It is evident that in a warmer climate, the air could hold more moisture, and thus, the thermodynamic effects amplify the intensity of PmE but do not alter its spatial pattern (Fig. 2(a); Held and Soden, 2006). Note that the hemispheric asymmetries of atmospheric energy could induce regional meridional circulation anomalies and thus alter the distribution of PmE anomalies during the mid-Pliocene via the $\delta MCD_{D\_M}$ term at low latitudes. The PlioPMIP2 ensemble simulations suggest that hemispheric asymmetries of atmospheric energy are the key factor altering the spatial pattern of PmE via changes in the local meridional circulation. However, we should note that a noticeable intermodel spread exists in capturing the main features in the past warm climate, particularly for the changes in Walker circulation, such as the large spread in the simulated changes in the intensity of PWC, dSST and dSLP in Fig. 10, consistent with previous studies (Oldeman et al., 2021). Further effort to understand the intermodel uncertainty needs to be explored in future work. In addition, previous studies indicate that the storm track (transient eddy component) may play a key role in changes in PmE for mid- to high latitudes (Seager et al., 2010; Han et al., 2019a; Han et al., 2019b). Due to the lack of hourly model data, we mainly discuss the relative contributions from moisture budget components to changes in PmE at low latitudes in this paper. Much more work should be conducted to study the impact of storm tracks on changes in PmE during the mid-Pliocene using hourly data in the future at mid-to-high latitudes.

   Note that the global temperature during mid-Pliocene is controlled by the combined effects of boundary conditions (e.g., $CO_2$ level, vegetation and topography) (Haywood et al., 2016). Any changes in each boundary condition could induce large-scale hydrological cycling changes. For example, the role of remote biophysical effects in the northern mid-high latitudes is highlighted in driving the variation of monsoon rainfall in low latitudes and shift of ITCZ, since the needleleaf tree expands greatly northward in eastern Eurasia during mid-Pliocene (Chase et al., 2000; Swann et al., 2014; Mahmood et al., 2014; Zhang and Jiang, 2014; Burls and Fedorov, 2017). In addition, some studies indicate that there exist uncertainties of boundary conditions of changing South Asian summer monsoon hydrological cycling. Sarathchandraprasad et al. (2017) indicate the tectonically induced reorganization of the Indonesian Throughflow can strengthen the SASM during mid-Pliocene due to the increased cross-equatorial pressure gradient. Recent study by Prescott et al. (2019) highlight the substantially influence of orbital forcing on the changes in SASM during mid-Pliocene. The simulations suggest that tectonic uplifts in South African Plateaus can strengthen the SASM as well (Zhang and Liu, 2013). Based on these studies, the boundary conditions applied by the PlioMIP2 models are important to hydrological cycling during mid-Pliocene in low latitudes. However, the relative impact of boundary conditions on hydrological cycling still remains uncertainties. In addition,

not all models carry out the sensitivity experiments designed in PlioMIP2, increasing the difficulty of exploring their relative contributions to PmE changes, and these questions need to be further explored in the future.

*Data availability.* To access the PlioMIP2 database, please send a request to Alan M. Haywood (a.m.haywood@leeds.ac.uk). PlioMIP2 data from CESM2, EC-Earth3-LR, GISS-E2-1-G, IPSL-CM6A-LR and NorESM1-F can be obtained from the Earth System Grid Federation (ESGF, 2020, https://esgf-node.llnl.gov/search/cmip6/). CCSM4 and CESM1.1 can be obtained from https://www.cesm.ucar.edu/models/. The reconstructed SST is from the alkenone-derived $U_{37}^{K'}$ index and foraminifera calcite Mg/Ca and can be accessed from https://pliovar.github.io/km5c.html.

*Author contributions.* Qiong Zhang and Zixuan Han designed the work, Zixuan Han wrote the manuscript under supervision from Qiong Zhang. Zixuan Han did the analyses and programming with the help of Jianbo Cheng and Qin Wen. All the other co-authors provided the PlioMIP2 model data and commented on the manuscript.

*Competing interests.* The authors declare that they have no conflict of interest.

*Special issue statement.* This article is part of the special issue "PlioMIP Phase 2: experimental design, implementation and scientifific results". It is not associated with a conference.

*Acknowledgements.* This research has been supported by the National Natural Science Foundation of China (Grant No. 42130610), the Swedish Research Council (Vetenskapsrådet, grant no. 2013-06476 and 2017-04232), the Fundamental Research Funds for the Central Universities (Grant No. B210201009) and the National Key R&D Program of China (Grant No. 2017YFC1502303). JC acknowledges financial support from the National Natural Science Foundation of China (Grant No. 42005012) and the Natural Science Foundation of Jiangsu Province (Grant No. BK20201058). QW acknowledges financial support from the National Natural Science Foundation of China (Grant No. 42106016) and the China Postdoctoral Science Foundation funded project (Grant No. 2021M691623). The model simulations with EC-Earth3 and data analysis were performed using ECMWF's computing and archive facilities and the Swedish National Infrastructure for Computing (SNIC) at the National Supercomputer Centre (NSC) partially funded by the Swedish Research Council through grant agreement no. 2018-05973. CJRW acknowledges the financial support of the UK Natural Environment Research Council funded SWEET project (Super-Warm Early Eocene Temperatures), research grant NE/P01903X/1. NJB acknowledges supported from the National Science Foundation (NSF; AGS-1844380 and OCN-2002448), as well as the Alfred P. Sloan Foundation as a Research Fellow. RF acknowledges the sponsorship by U.S. National Science Foundation grants 1903650 and 1814029. The contributions of BLO-B, ECB, and NR are based upon work supported by the National Center for Atmospheric Research, which is a major facility sponsored by the NSF under Cooperative Agreement No. 1852977. The CESM project is supported primarily by the National Science Foundation (NSF). Computing and data storage resources for the CESM and CCSM4 simulations, including the Cheyenne supercomputer (doi:10.5065/D6RX99HX), were provided by the Computational and Information Systems Laboratory (CISL) at NCAR. XL acknowledges financial support from the National Natural Science Foundation of China (NSFC, grant no. 42005042) and China Scholarship Council (201804910023). The NorESM simulations benefitted from resources provided by UNINETT Sigma2–the National Infrastructure for High Performance Computing and Data Storage in Norway. The work by ASvdH and MLJB was carried out under the program of the Netherlands Earth System Science Centre (NESSC), financially supported by the Ministry of Education, Culture and Science (OCW grant no. 024.002.001). Simulations with CCSM4-Utrecht were performed at the SURFsara Dutch national computing facilities and were sponsored by NWO-EW (Netherlands Organisation for Scientific Research, Exact Sciences) (Project no. 17189 and 2020.022). CS and GL acknowledge computational resources from the Computing and Data Centre

of the Alfred-Wegener-Institute ― Helmholtz-Centre for Polar and Marine Research. CS and GL acknowledge funding via the Helmholtz Climate Initiative REKLIM and the Alfred Wegener Institute's research programme "Changing Earth-Sustaining our Future". The PRISM4 reconstruction and boundary conditions used in PlioMIP2 were funded by the U.S. Geological Survey Climate and Land Use Change Research and Development Program. Any use of trade, firm, or product names is for descriptive purposes only and does not imply endorsement by the U.S. Government.

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
