# Peer review of "Evaluating the large-scale hydrological cycle response within the PlioMIP2 ensemble"

_Climate of the Past, 2021_

## Referee Comment (RC2)

**Review of "Evaluating the large-scale hydrological cycle response within the PlioMIP2 ensemble"**

by Zixuan Han et al.

**Manuscript No:** CP-2021-72

**Recommendation:** Major revisions

**General Comments:** In this article the authors investigated the simulated changes in the mid-Pliocene Precipitation - Evaporation (P - E) pattern relative to the PI simulation. In addition, the authors also attempted to attribute these hydrological cycle changes to its dynamic and thermodynamic component, which is partially influenced by Seager et al. (2010). The results in this article could be useful for a detailed understanding of the mid-Pliocene hydro-climate but in its current form it is lacking some clarifications (listed below) that must be addressed before the publication.

**Specific Comments**

- Equation (1) only deals with the thermodynamic and mean circulation dynamics contribution. The transient eddy contribution and the surface quantities are included in the residual (R) term. It will be nice to see how different is the MMM PmE from the sum of TH and MCD component. This is since the poleward moistening is believed to be a result from the transient eddies and the difference might show this feature. I would also suggest to extend the meridional domain from 60° to 90°.

- Line 58: The wet-region-getting-wetter and dry-regions-getting drier phenomenon is primarily valid over the ocean. Over the landmass this characteristic is not always true. A study by Greve et al. (2014) reported that only 10.8% of the global land area shows dry gets drier and wet gets wetter pattern.

- Line 210-211: Did not understood properly the meaning of strengthened East Asian summer monsoon circulation (is it more rainfall or strong winds?) and how it affects the MMM PmE changes over Southeast Asia.

- Line 253-254: "mid-high latitudes" is a very broad region. You might specify the areas dominated by the climate mean horizontal circulation e.g. Western coast of South America, Southern tip of South Africa, a region extending from the Southern tip of South America to the Central tropical Pacific Ocean.

- Line 274: What does it mean by Southern part and Northern part of the deep tropics? Is it Southern and Northern hemispheric part of the deep tropics or something else?

- Line 275: It is not clear to me how one can infer from Fig. 4(f) that ITCZ has shifted northward.

- Line 277-278: Did not understood the meaning of the sentence starting with " There is a tendency...".

- Line 327-328: Unable to agree with this statement. The changes in the MMM SST (Fig. 6a) and specific humidity (Fig. 6b) does not seemed to be in agreement. The MMM SST changes are largest over the higher latitudes and in contrary the largest changes in the specific humidity occurs over the tropics. Also you need to change the figure reference (it is not 5b, it should be 6b).

**Technical comments**

- Line 68: remove enhance or increase. Both have the same meaning.

- Line 214: I think the figure referencing is wrong here. It should be Fig. 6b.

- Line 220: It would be nice to add one more plot here which will represent the changes in the MMM PmE due to the combined TH and MCD component.

- Line 253: "... unaltered divergence of the mean zonal circulation.." puzzled a bit with the usage of "unaltered" and "mean". I believe you can remove the "unaltered".

- Line 269: "Overwhelming" is not the correct word in this context. Probably one can use "dominating".

- Line 317-318: In Figure 5 caption the orange band marking is wrong and I don't see any green band.

- Please make the contour intervals in such a way that it removes the zero contour lines. Applicable to all the figures. Often inclusion of the zero contours lead to misinterpretation of the results.

- I think stippling should be used on the non-significant regions so that we have a clear view of the significant results.

**Reference**

Greve, P., Orlowsky, B., Mueller, B., Sheffield, J., Reichstein, M., & Seneviratne, S. I. (2014). Global assessment of trends in wetting and drying over land. Nature geoscience, 7(10), 716-721.

---

## Author Comment (AC1)

**Author response to Reviewer #1**

The authors thank the reviewers and editors for beneficial and helpful suggestions for this manuscript. We have carefully revised the manuscript according to the suggestions. According to the reviewer's comments, the following replies have been made in blue.

**-----**

**Reviewer's Comments to Authors:**

This paper evaluates the contribution of thermodynamical and dynamical effects to changes in P minus E in the simulated Pliocene warming. It decomposes the moisture budget into thermodynamical and dynamical contribution, which are further decomposed by applying the 3-pattern decomposition of the atmospheric circulation. The authors find that both mechanisms have significant contributions in explaining changes in PmE. In general, they find that thermodynamical effect can explain the wetter tropics and drier subtropics, however, when looking to more regional changes within the tropics and subtropics, dynamical effects have an important contribution. The paper goes on to explore possible reasons for the changes in the moisture budget components.

I think that the methods applied to achieve the main objective are appropriate and that the paper shows an interesting contribution to the research in the field. Therefore, I recommend the publication of this manuscript by the journal Climate of the Past, but I see some major issues that need to be addressed before publication.

**General comments:**

1. The main objective of the is study is to evaluate the relative contribution of thermodynamical and dynamical effects in the PlioMIP models. However, the paper does not compare the relative contributions of the two. The authors correctly applied the moisture budget decomposition, but they do not quantitively compare how much of each of the components contributes to the total PmE pattern. I think this analysis is crucial to this evaluation and would greatly improve the manuscript.

**Response:** Thanks for the comments. To quantitively compare the contribution of the dynamic and thermodynamic components to the total PmE change, we firstly calculate the zonal mean of moisture budget components as shown in supplement Figure 1a. The changes in PmE generally show a wet-regions-getting-wetter and dry-regions-getting-drier pattern during the mid-Pliocene, and this is mainly contributed by the thermodynamic term. Furthermore, the thermodynamic term leads to increase in PmE by ~58.6% over tropic region, and decrease in PmE by ~84.6% over subtropics, respectively. The dynamic effect has relative smaller contribution, with increase in PmE by  $\sim$ 7.9% in tropic and  $\sim$ 10.5% in subtropics. In fact, the  $\delta$ TH component does not describe the full contribution of the changes in PmE everywhere. For example, over the North Indian Ocean, the  $\delta MCD_{DZ}$  (decomposed by the  $\delta MCD$ term) is the first-order contribution to strengthen the dynamic effect (by ~45%) and hence enhance the PmE over north Indian ocean (Figure 4g), which is caused by the shift of Walker circulation (Figure 10), the contribution of thermodynamic is negligible. We show the contribution of each moisture budget components over north Indian ocean in supplementary Figure 2.

**We have added related sentences in the new version as follows:**

"In general, the thermodynamic term increases PmE by  $\sim$ 58.6% over tropic region, and decrease in PmE by  $\sim$ 84.6% over subtropics (not shown), respectively." (See Line 248-249)

"..., the convergence of zonal circulation anomalies ( $\delta MCD_{D-Z}$ ) is the first-order

contribution to strengthen the dynamic effect (by  $\sim$ 45%) and hence enhance the PmE (Fig.4(g))." (See Line 296-298)

Figure S1. (a) The zonal average of the moisture budget components changes for PlioMIP2 models. (b) The average of moisture budget components in tropics ( $5^{\circ}$ S-15°N) and subtropics ( $30^{\circ}$ -5°S and 15°-30°N). Units: mm/day.

Figure S2. The average of moisture budget components over north Indian ocean (0-20°N, 40°-100°E). unit: mm/day.

2. The authors have produced interesting results that should be introduced more carefully. There is a contradiction in the abstract. The authors initially exemplify the changes in PmE over the Pacific ITCZ as a result of the thermodynamical effect. And a few lines below, they evaluate that the ITCZ shifts 1° northward (which is a result of dynamical changes). To me, the results show that in a very general (large-scale) manner the thermodynamical effects explain the wetter deep tropics and drier subtropics. But when looking to regional changes within the tropics and subtropics the

dynamical effect plays an important role (i.e. northward ITCZ shift, southward SPCZ shift, wetter north Indian ocean).

**Response:** The reviewer's suggestions are very beneficial. The reviewer's opinion is right. Indeed, our results indicate that the thermodynamical effects explain the wetter deep tropics and drier subtropics in a large-scale manner. These results can also be found in Figure S1 (see comment 1). Although the dynamic term is relative smaller than thermodynamic term in a general manner (Figure S1), the dynamic effects can not be neglect when looking to regional changes in the low latitudes. **To avoid the confusion in abstract, we have revised and added the related sentences as follows:**

"Note that the dynamic effect play a more important role in regional PmE changes (i.e., northward ITCZ shift and wetter in north Indian ocean)." (See Line 44-45).

3. It was not clear to me what boundary conditions the PlioMIP2 models have applied. For example, did they apply 400 ppm CO2? Are there changes in the extension of the ice sheets in both hemispheres? Vegetation? Changes in the configuration of the continents? Also, I think there should have a final discussion as how these boundary conditions could have affected the simulated PmE changes.

**Response:** Thanks for the comments. We use a suite of model simulations from 13 the Pliocene Model Intercomparison Project Phase 2 (PlioMIP2) participating the Coupled Model Intercomparison Project phase 6 (CMIP6) (Haywood et al., 2016). Boundary conditions for mid-Pliocene simulations are derived from the latest iteration of the U.S. Geological Survey PRISM data set (PRISM4) (Dowsett et al., 2016), including land versus sea, vegetation, soils, lakes, land ice cover, topography and bathymetry. The concentration of atmospheric  $CO_2$  level for mid-Pliocene and PI simulations are prescribed at 400ppm and 280ppm, respectively. Other solar output, trace gases and orbital parameters for mid-Pliocene simulations are set to be consistent with each model's PI simulation. Modeling groups are given the option to either prescribe vegetation changes or simulate vegetation changes using a dynamic global vegetation model. And only one modeling group (COSMOS) in the suite of simulations we analyzed opted to use a dynamic configuration for vegetation.

We agree with the reviewer that it is important to discuss the impact of boundary conditions on changes in PmE during mid-Pliocene. Note that the large-scale atmospheric circulation changes are related to the temperature pattern anomalies. Earlier studies demonstrate that the temperature changes during mid-Pliocene are controlled by the combined effects of external forcings, including the atmospheric  $CO_2$  concentration, land cover (surface vegetation and land ice cover), and topography (Haywood et al., 2016). Any changes in each boundary conditions could lead to changes in PmE through dynamic (i.e., atmospheric circulation changes) or thermodynamic (i.e., atmospheric moisture changes) effects. For example, several studies highlight the role of remote biophysical effects in the northern mid-high latitudes in driving the variation of monsoon rainfall in low latitudes, which could lead to land surface characteristics changes (i.e., evapotranspiration and albedo),

altering shifts of ITCZ due to the imbalance of atmospheric energy budget and hence changing monsoon rainfall (Swann et al., 2014; Chase et al., 2000; Mahmood et al., 2014). Proxy and simulations suggest that the needleleaf tree shifts greatly and expands northward in eastern Eurasia during mid-Pliocene (Zhang and Jiang, 2017; Feng et al., 2021). These changes in land cover during mid-Pliocene may alter PmE in the low latitudes as well, especially in monsoon regions. In addition, some studies indicate that there exist uncertainties of boundary conditions of changing South Asian summer monsoon hydrological cycling. Sarathchandraprasad et al. (2017) indicate the tectonically induced reorganization of the Indonesian Throughflow can strengthen the South Asian summer monsoon (SASM) during mid-Pliocene due to the increased cross-equatorial pressure gradient. Recent study by Prescott et al., (2019) highlight the substantially influence of orbital forcing on the changes in SASM during mid-Pliocene. The simulations suggest that tectonic uplifts in South African Plateaus can strengthen the SASM as well (Zhang and Liu, et al., 2013). Based on these studies, the boundary conditions applied by the PlioMIP2 models are important to hydrological cycling during mid-Pliocene in low latitudes. However, since not all models carry out the sensitivity experiments designed in PlioMIP2, it remains difficult to distinguish which change in boundary conditions is more dominant for the PmE changes in low latitudes.

We have added the related description of boundary conditions and discussion of the possible effects in the new version as follows:

"..., including soils, lakes, land ice cover, vegetation, topography, bathymetry. The CO2 level for mid-Pliocene and PI simulations are set at 400ppm and 280ppm, respectively." (See line 104-106)

"Zhang et al. (2016) indicate that the combined influence of SST and  $CO_2$  level, as well as the vegetation changes, play very important role in changing the atmospheric circulation over North Africa during mid-Pliocene, owing to the increased net atmospheric energy there." (See line 212-214)

"Note that the global temperature during mid-Pliocene is controlled by the combined effects of boundary conditions (e.g., CO2 level, vegetation and topography) (Haywood et al., 2016). Any changes in each boundary conditions could induce large-scale hydrological cycling changes. For example, poxy and simulations suggest that the needleleaf tree expands greatly 13rthward in eastern Eurasia during mid-Pliocene (Zhang and Jiang, 2014; Burls and Fedorov, 2017). Some studies highlight the role of remote biophysical effects in the northern mid-high latitudes in driving the variation of monsoon rainfall in low latitudes, which could lead to the shift of ITCZ due to the imbalance of atmospheric energy budget and hence changing monsoon rainfall (Swann et al., 2014; Chase et al., 2000; Mahmood et al., 2014). Since the needleleaf tree expands greatly northward in eastern Eurasia during mid-Pliocene (Zhang and Jiang, 2014; Burls and Fedorov, 2017), this changes may remarkably influence the changes in PmE pattern in low latitudes. In addition, some studies indicate that there exist uncertainties of boundary conditions of changing South Asian summer monsoon hydrological cycling. Sarathchandraprasad et al. (2017) indicate the tectonically induced reorganization of the Indonesian Throughflow can

strengthen the SASM during mid-Pliocene due to the increased cross-equatorial pressure gradient. Recent study by Prescott et al., (2019) highlight the substantially influence of orbital forcing on the changes in SASM during mid-Pliocene. The simulations suggest that tectonic uplifts in South African Plateaus can strengthen the SASM as well (Zhang and Liu et al., 2013). Based on these studies, the boundary conditions applied by the PlioMIP2 models are important to hydrological cycling during mid-Pliocene in low latitudes. However, the relative impact of boundary conditions on hydrological cycling still remain uncertainties. In addition, since not all models carry out the sensitivity experiments in PlioMIP2, it increase the difficulty of exploring their relative contributions to PmE changes, and these questions need to be further explored in the future" (See Line 504-530)

**References:**

Haywood, A. M., Dowsett, H. J., Dolan, A. M., Rowley, D., Abe-Ouchi, A., Otto-Bliesner, B., Chandler, M. A., Hunter, S. J., Lunt, D. J., Pound, M., and Salzmann, U.: The Pliocene Model Intercomparison Project (PlioMIP) phase 2: scientific objectives and experimental design, Clim. Past, 12, 663-675, https://doi.org/10.5194/cp-12-663-2016, 2016.

Zhang R. and Jiang D. 2014. Impact of vegetation feedback on the mid-Pliocene warm climate. Advances in Atmospheric Sciences, 31: 1407-1416.

Swann, A.L., Fung, I.Y., Liu, Y. and Chiang, J.C., 2014. Remote vegetation feedbacks and the mid-Holocene green Sahara. Journal of climate, 27(13), pp.4857-4870.

Chase, T.N., Pielke Sr, R.A., Kittel, T.G.F., Nemani, R.R. and Running, S.W., 2000. Simulated impacts of historical land cover changes on global climate in northern winter. Climate dynamics, 16(2), pp.93-105.

Mahmood, R., Pielke Sr, R.A., Hubbard, K.G., Niyogi, D., Dirmeyer, P.A., McAlpine, C., Carleton, A.M., Hale, R., Gameda, S., Beltrán-Przekurat, A. and Baker, B., 2014. Land cover changes and their biogeophysical effects on climate. International journal of climatology, 34(4), pp.929-953.

Burls, N. J. and Fedorov, A. V.: Wetter subtropics in a warmer world: contrasting past and future hydrological cycles, Proc. Natl. Acad. Sci. U. S. A., 114, 12888, https://doi.org/10.1073/pnas.1703421114, 2017.

Sarathchandraprasad, T., Tiwari, M. and Behera, P., 2021. South Asian Summer Monsoon precipitation variability during late Pliocene: Role of Indonesian Throughflow. Palaeogeography, Palaeoclimatology, Palaeoecology, 574, p.110447.

Zhang R. and Liu X.D., 2010. The effects of tectonic uplift on the evolution of Asian summer monsoon climate since Pliocene. Chinese Journal of Geophysics, 53(6), pp.948-960.

4. Section 3 is an important section of paper that introduces the simulated PmE changes that will be further explored in the next sections and should be carefully revised. To me the key message of section 3 is that, in a first look to changes in PmE, there seem to have important contributions from both TH and MCD, but the authors

mainly discuss TH (see comment 2 and detailed comments below).

**Response:** Thanks for the comments. We have added a few sentences to try to avoid this confusion as follows:

"Note that there is moistening signal in SPCZ's southern part in the tropical southern Pacific." (See line 204-205)

"Earlier studies indicate these features of changes in PmE at low latitudes are linked to the increased specific humidity (i.e., changes in thermodynamic effect). However, there are some opposite phenomenon as well, when looking to the regional changes in PmE (i.e., north Inidan Ocean, North Africa and SPCZ). These may suggest that another factor, such as atmospheric circulation anomalies (i.e., changes in dynamic effect), may play an important role in changing regional PmE pattern at low latitudes." (See Line 227-231)

5. Section 5.1: it is interesting to examine the global pattern in wind divergence and humidity. But SST pattern must be analysed more thoroughly, especially in the tropics where local SST will probably help to explain the thermodynamical effect.

Xie, S.-P., C. Deser, G. A. Vecchi, J. Ma, H. Teng, and A. T. Wittenberg, 2010: Global warming pattern formation: Sea surface temperature and rainfall. J. Climate, 23, 966–986, https://doi.org/10.1175/2009jcli3329.1.

Response: The reviewer's suggestions are very beneficial. Figure S3 shows the changes in annual mean SST and wind at 850 hPa. The SST warming is amplified in the northwest tropical Indian Ocean, whereas it is reduced off the Indonesian coast, showing tropical Indian Ocean dipole (IOD)-like pattern. The sharp SST gradients drive strong southeasterly wind anomalies (Figure S3) over the equator. Figure S4 show the changes in annual mean of zonal wind (U) versus changes in SST for 13 PlioMIP2 models. These results can also be seen in Figure S4. As expected, the models with an enhanced zonal SST gradient across the equatorial Indian Ocean tend to produce increased easterly wind anomalies, with the inter-model correlations of -0.9. Xie et al. (2010) suggest that this easterly wind anomalies may shoal the thermocline in the east, helping cool SST there via upwelling. This indicates that this SST anomalies over tropical Indian Ocean may be related to the Bjerknes feedback. Studies indicate that the simulated North Atlantic warming is related to an intensified mid-Pliocene AMOC (Li et al., 2020). However, Zhang et al. (2021) suggest that the increased background ocean vertical mixing parameters can be also responsible for the warm SSTs there. Note that the relative cool SST in Southeast Pacific and Atlantic, which is collocated with the intensified southeast trades, suggesting the role of WES feedback (Xie et al., 2010).

**According to the comments, we have added more discussion of the SST pattern as follows:**

"Note that the SST warming is amplified in the northwest tropical Indian Ocean, whereas it is reduced off the Indonesian coast, showing tropical Indian Ocean dipole (IOD)-like pattern. The sharp SST gradients drive strong southeasterly wind anomalies on the equator (not shown). Xie et al. (2010) suggest that this easterly wind anomalies may shoal the thermocline in the east, helping cool SST there via upwelling,

and indicating this SST anomalies over tropical Indian Ocean may be related to the Bjerknes feedback. The simulated North Atlantic warming is might related to an intensified mid-Pliocene AMOC (Li et al., 2020). However, Zhang et al. (2021) suggest that the increased background ocean vertical mixing parameters can be also responsible for the warm SSTs there. However, the relative cooler exist in Southeast Pacific and Atlantic, which is collocated with the intensified southeast trades, suggesting the role of wind-evaporation-SST feedback (Xie et al., 2010). These SST warming patterns are consistent with current studies (Haywood et al., 2020; Williams et al., 2021)." (See line 345-355)

**References:**

Xie, S.-P., C. Deser, G. A. Vecchi, J. Ma, H. Teng, and A. T. Wittenberg, 2010: Global warming pattern formation: Sea surface temperature and rainfall. J. Climate, 23, 966–986, https://doi.org/10.1175/2009jcli3329.1.

Li, X., Guo, C., Zhang, Z., Otterå, O.H. and Zhang, R., 2020. PlioMIP2 simulations with NorESM-L and NorESM1-F. Climate of the Past, 16(1), pp.183-197.

Zhang, Z., Li, X., Guo, C., Otterå, O.H., Nisancioglu, K.H., Tan, N., Contoux, C., Ramstein, G., Feng, R., Otto-Bliesner, B.L. and Brady, E., 2021. Mid-Pliocene Atlantic Meridional Overturning Circulation simulated in PlioMIP2. Climate of the Past, 17(1), pp.529-543.

---

## Author Comment (AC2)

**Author response to Reviewer #2**

**The authors thank the reviewers and editors for beneficial and helpful suggestions for this manuscript. We have carefully revised the manuscript according to the suggestions. According to the reviewer's comments, the following replies have been made in blue.**
* * *
**General Comments:** In this article the authors investigated the simulated changes in the mid-Pliocene Precipitation-Evaporation (P-E) pattern relative to the PI simulation. In addition, the authors also attempted to attribute these hydrological cycle changes to its dynamic and thermodynamic component, which is partially influenced by Seager et al. (2010). The results in this article could be useful for a detailed understanding of the mid-Pliocene hydro-climate but in its current form it is lacking some clarifications (listed below) that must be addressed before the publication.

**Specific comments:**

**1. Equation (1) only deals with the thermodynamic and mean circulation dynamics contribution. The transient eddy contribution and the surface quantities are included in the residual (R) term. It will be nice to see how different is the MMM PmE from the sum of TH and MCD component. This is since the poleward moistening is believed to be a result from the transient eddies and the difference might show this feature. I would also suggest to extend the meridional domain from 60◦ to 90◦.**

**Response:** Thanks for the comments. In fact, one of our main points in this paper is to distinguish the relative contribution of changes in Hadley circulation and Walker circulation to changes of PmE during the mid-Pliocene. Since these circulation anomalies mainly happen at the low latitudes, thus the transient eddies at mid-to-high latitudes are not discussed in this paper. And we have highlighted this point in the paper. For example, in the abstract, we have indicated that "Here, we apply a moisture budget analysis to investigate the response of the large-scale hydrological cycle **at low latitudes** within a 13-model ensemble from the Pliocene Model Intercomparison Project Phase 2 (PlioMIP2)." In the introduction, we have shown "However, it is difficult to distinguish the relative impact of the Hadley circulation and Walker circulation on Pliocene hydrological cycling **at low latitudes**. Fortunately, the three-pattern decomposition of global atmospheric circulation (3P-DGAC; Hu et al., 2017, 2018b, c) method can help us to decompose atmospheric circulation into zonal (i.e., local Walker circulation) and meridional (i.e., local Hadley circulation) circulation **at low latitudes.**"

Indeed, the review's comments are right, the contribution of transient eddies is quite important for poleward moisture. Previous studies indicate that the storm track (transient eddy component) may play a key role in changes in PmE for mid- to high latitudes (Seager et al., 2010; Han et al., 2019a; Han et al., 2019b). In addition, the residual term includes transient eddy contribution, the surface quantities, the nonlinear term and model biases. In the Section 6 "Conclusion and discussion" we have mentioned this, that is "Due to the lack of hourly model data, we mainly discuss the relative contributions from moisture budget components to changes in PmE at low

latitudes in this paper. Much more work should be conducted to study the impact of storm tracks on changes in PmE during the mid-Pliocene using hourly data in the future at mid-to-high latitudes."

Therefore, since discussion of transient eddies at mid-high latitudes are not our main point in this paper and there still remain difficulties to distinguish this term from the residual term, and we decide to not discuss this topic in this paper, but in the future studies.

**2. Line 58: The wet-region-getting-wetter and dry-regions-getting drier phenomenon is primarily valid over the ocean. Over the landmass this characteristic is not always true. A study by Greve et al. (2014) reported that only 10.8% of the global land area shows dry gets drier and wet gets wetter pattern.**

**Response:** Thanks for the comments. We have updated the descrption of changes over the ocean and land in the new version as follows:

"Note that this phenomenon is primarily focus on the ocean. A study by Greve et al. (2014) report that only 10.8% of the global land area shows dry gets drier and wet gets wetter pattern." (See Line 62-63)

Reference:

Greve, P., Orlowsky, B., Mueller, B., Sheffield, J., Reichstein, M., and Seneviratne, S.I. 2014. Global assessment of trends in wetting and drying over land. Nature geoscience, 7(10), 716-721.

**3. Line 210-211: Did not understood properly the meaning of strengthened East Asian summer monsoon circulation (is it more rainfall or strong winds?) and how it affects the MMM PmE changes over Southeast Asia.**

**Response:** Thanks for the comments. Here the strengthened East Asian summer monsoon *circulation* means strong winds, and subsequent more rainfall as explained in the following. Abundant studies have indicated that the monsoon rainfall changes are highly related to the large-scale circulation anomalies. To examine the changes in monsoon circulation and precipitation, we further show the climate mean and changes in precipitation and 850hPa wind field in Figure S1. Compared to the PI simulation (Figure S1a), there is a robust increased South Asian summer (SAS) rainfall revealed by PlioMIP2 ensembles, with precipitation maxima over north-eastern part of Indian subcontinent and south of Western Ghats region (Figure S1b). These changes in SAS rainfall are consistent with proxy indicators of precipitation over northern ISM region (Feng et al., 2021). Concurrently, the change in 850-hPa wind shows pronounced westerly winds anomalies over SAS region (Figure S1b), indicating strengthened ISM circulation. The similar results can also been seen over East Asian summer (EAS) monsoon region. There is a remarkably strengthened southerly winds anomalies over EAS region (Figure S1b), favoring to increase rainfall there. Therefore, this increased precipitation in monsoon regions result in the positive precipitation minus evaporation (PmE) there.

[Figure]

Figure S1. Ensemble (a) climatologies and (b) changes of precipitation (shading, mm/day) and 850 hPa wind (vector, m/s) during boreal summer season (June to September), simulated in the 13 PlioMIP2 models. The solid box marks the region (65°-95°E; 10°-30°N) and (105°-130°E; 20°-33°N) in which denote the South Asian summer monsoon and East Asian summer monsoon region, respectively. The red stippling in (b) indicates regions where at least 10 simulations show the same sign.

**4. Line 253-254: "mid-high latitudes" is a very broad region. You might specify the areas dominated by the climate mean horizontal circulation e.g. Western coast of South America, Southern tip of South Africa, a region extending from the Southern tip of South America to the Central tropical Pacific Ocean.**

**Response:** The reviewer's comments are right. We have specified the areas dominated by the climate mean horizontal circulation in the new version as follows:

"At mid-high latitudes, the δTH induced by climate mean horizontal circulation is caused by changes in moisture advection at mid-high latitudes (Fig. 3(k)), e.g., Western coast of North America, a region extending from the Southern tip of South America to the Central tropical Pacific Ocean, and Southern tip of South Africa." (See Line 266-269)

**5. Line 274: What does it mean by Southern part and Northern part of the deep tropics? Is it Southern and Northern hemispheric part of the deep tropics or something else?**

**Response:** Thanks for you comments. We have changed the confusion sentence in the new version as follows:

"... appears to dry the deep tropics but moisten its Northern hemispheric part of the deep tropics..." (See Line 289)

**6. Line 275: It is not clear to me how one can infer from Fig. 4(f) that ITCZ has**

shifted northward.

**Response:** Thanks for your comments. In Figure 8c, our studies indicate the northward shift of ITCZ, which could reorganize the atmospheric circulation (i.e., Hadley circulation). From the perspective of decomposition of moisture budget equation, the $\delta MCD_{D\_M}$ term reflect the changes in local Hadley circulation. As shown in Figure 4f, this term appears to dry the deep tropics but moisten its Northern hemispheric part of the deep tropics. We further calculate the relationship between the precipitation averaged over the Northern hemispheric part of the deep tropics (10°N-15°N, 180°E-180°W) in Figure S2. Indeed, the change in precipitation over the Northern hemispheric part of the deep tropics is related to the shift of ITCZ. Models with a northward shift of ITCZ tend to produce more rainfall over the Northern hemispheric part of the deep tropics, with the inter-model correlations of 0.7. Statistically, the inter-model correlation coefficients exceed the 0.01 significance level according to the t test.

Therefore, we highlight that the positive contribution of $\delta MCD_{D\_M}$ term over the Northern hemispheric part of the deep tropics reflect the impact of shift of ITCZ.

[Figure]

Figure S2. Changes in the latitudes of the center of annual mean precipitation between 20°S and 20°N (reflecting the shift of ITCZ; unit: °) versus changes in precipitation (units: mm/day) averaged over the Northern hemispheric part of the deep tropics (10°N-15°N).

**7. Line 277-278: Did not understand the meaning of the sentence starting with "There is a tendency...".**

**Response:** Thanks for the comments. We have revised the confusion sentence in the new version as follows:

" $\delta MCD_{D\_M}$ term contributes to reduce PmE over SPCZ region, but this term increase PmE over the southern part of SPCZ in the tropical southern Pacific." (See Line 292-293)

**8. Line 327-328: Unable to agree with this statement. The changes in the MMM SST (Fig. 6a) and specific humidity (Fig. 6b) does not seem to be in agreement. The MMM SST changes are largest over the higher latitudes and in contrary the largest changes in the specific humidity occurs over the tropics. Also you need to change the figure reference (it is not 5b, it should be 6b).**

Response: Thanks for the comments. To avoid the confusion, we have changed the statement in the new version as follows:

"As expected, the specific humidity is increased in the low-level troposphere in a warmer climate (Fig. 7(b)) (Murray, 1966; Held and Soden, 2006)" (See Line 354-356)

Actually, the specific humidity anomalies are related to the changes in SST. To explain why the SST warming is larger at high latitudes than the low latitudes (Figure 7a) but with much less specific humidity increase there (Figure 7b), we further discuss the relationship between SST and specific humidity as follows.

Starting from the Clausius-Clapeyron (C-C) equation derived from thermodynamic theories, we can get the relationship between saturated vapor pressure ($e_s$) and temperature ($t$) as follows:

$$\frac{de_s}{dT} = \frac{L_v e_s}{R_v T^2} \qquad (1)$$

Here, $T$ is temperature, $e_s$ is the saturated vapor pressure with respect to a flat liquid surface. $R_v$ is the specific gas constant. $L_v$ is the latent heat of vaporization (assuming $L_v$ is constant). We can get the simple Tetens's empirical formula (Murray et al., 1966) to calculate the $e_s$.

$$e_s = 6.1078 exp[\frac{17.2693882(T-273.16)}{T-35.86}] \qquad (2)$$

Based on the equation (2), we calculate the changes in es follows the response of $t$ in Figure S2. Obviously, the changes in $e_s$ is the function of T with logarithm to base $e$. Note that the tropical SST is more than 20℃, however, SST is reduced rapidly less than 10℃ on average at high latitudes. If we assume the SST change is $\Delta t$ both at low latitudes and high latitudes, the change in $e_s$ is $\Delta e_{s2}$ and $\Delta e_{s1}$ (as shown in Figure S3), respectively. **Noticeably, the $\Delta e_{s2}$ is much larger than $\Delta e_{s1}$, indicating the $e_s$ anomaly is more sensitive to the SST changes at the low latitudes (pronounced warm region) than the high latitudes (relative cooler region).**

Additionally, the relationship between relative humidity ($U_w$) and specific humidity ($q$) can be write as follow:

$$U_w \approx \frac{q}{q_s} \qquad (3)$$

Then, we can get the relationship between $q$ and $e_s$:

$$q \approx U_w q_s \approx \frac{U_w \varepsilon}{p} e_s(T) \approx A e_s(T) \qquad (4)$$

Here, $U_w$ is relative humidity, $\varepsilon$ is 0.622, $p$ is air pressure. Thus, the results of $q$ are similar to $e_s(T)$ as mentioned above. That is, the specific humidity anomaly is much more sensitive to the SST changes at the low latitudes than the high latitudes.

This indicates that even SST warming is larger at high latitudes than the low latitudes, however, much less specific humidity increase for the former than the latter. **These results suggest the SST changes (Figure 7a) and specific humidity (Figure 7b) are reasonable.**

[Figure]

Figure S3. The changes in saturated vapor pressure ($e_s$; hPa) follows the response of temperature changes ($t$; ℃).

Reference:

Murray, F. W.. On the computation of saturation vapor pressure. Rand Corp Santa Monica Calif. 1966.

Held, I. M. and Soden, B. J.: Robust responses of the hydrological cycle to global warming, J. Clim., 19, 5686–5699, https://doi.org/10.1175/JCLI3990.1, 2006.

**Technical corrections:**

**9. Line 68: remove enhance or increase. Both have the same meaning.**

**Response:** Thanks for your carefully reading. We have removed the word "enhance the" in this sentence (See Line 68).

**10. Line 214: I think the figure referencing is wrong here. It should be Fig. 6b.**

**Response:** Thanks for your comment. We have changed this figure referencing in the new version.

**11. Line 220: It would be nice to add one more plot here which will represent the changes in the MMM PmE due to the combined TH and MCD component.**

**Response:** Thanks for your comment. We have added one more plot as Figure 5 to present the changes the MMM PmE due to the combined δTH and δMCD component. We have added the related sentence in the new version as follows:

"In summary, the dynamic and thermodynamic can explain the largest changes in

PmE at low latitudes (Figs. 5 vs 1)"   (See Line 310)

"Even the dynamic term overwhelmingly contributes to the increased changes in PmE over North African and Southeast Asian monsoon regions and the North Indian Ocean (Fig. 4 vs 5)." (See Line 313-315)

[Figure]

Figure 5. The estimated annual mean changes in the PmE (calculated by the sum of the dynamic term and thermodynamic term) of the mid-Pliocene minus PI control of the PlioMIP2 multimodel mean. Unit: mm day⁻¹.

**12. Line 253: "... unaltered divergence of the mean zonal circulation.." puzzled a bit with the usage of "unaltered" and "mean". I believe you can remove the "unaltered".**
**Response:** Thanks for your comment. We have removed "unaltered" in this sentence.

**13. Line 269: "Overwhelming" is not the correct word in this context. Probably one can use "dominating".**
**Response:** Thanks for your comment. We have changed the word "overwhelming" to "dominating" in new version.

**14. Line 317-318: In Figure 5 caption the orange band marking is wrong and I don't see any green band.**
**Response:** Thanks for your comment. We have changed the wrong Figure 5 caption in the new version as follows:

"Here, the tropical region is defined as the region between 10°S and 10°N (marked as an orange band), while the subtropical region refers to 10-30° N and 10-30° S (marked as an cyan band)." (See Line 336-338)

**15. Please make the contour intervals in such a way that it removes the zero contour lines. Applicable to all the figures. Often inclusion of the zero contours lead to misinterpretation of the results.**
**Response:** Thanks for your comment. We have added the Figures' labels of contour lines in the new version.

[Figure]

Figure 1: (a) Changes in multimodel mean (MMM) PmE for the mid-Pliocene compared with the PI simulation (shading), overlaid by the climatological MMM PmE of the PI simulation (for the contours, a solid line indicates positive values and a dashed line indicates negative values). The red solid curves represent the zero value. (b) The zonal average of the change in PmE, where the shading indicates the interquatile range among models. Stippling (left) indicates regions where at least 10 of 12 simulations in the model group agree on the sign of the ensemble mean. Units: mm·day⁻¹.

[Figure]

Figure 7: Change in (a) MMM SST (shading, unit: °C), (b) specific humidity (shading, unit: kg·kg⁻¹) overlaid by its climate mean for PI simulation. (c) and (d) show the MMM divergence of meridional $\vec{V}_M$ and zonal wind $\vec{V}_Z$ fields decomposed from the 3P-DGAC method at the 850 hPa level for PI simulation (unit: 10⁻⁶ s⁻¹). The circles in (a) are the anomalies of reconstructed SST from the alkenone-derived $U_{37}^{K'}$ index (Prahl and Wakeham, 1987) and foraminifera calcite Mg/Ca (Delaney et al., 1985). Stippling in (b-d) indicates regions where at least 10 of 13 simulations in the model group agree on the sign of the ensemble mean.

[Figure]

Figure 8: (a) Changes in annual mean MSF (shading; units: $10^{10}$ kg·s$^{-1}$) of meridional circulation for mid-Pliocene with respect to the PI simulation, overlaid by the climate mean MSF for the PI simulation (contours). The meridional wind $\vec{V}_M$ is decomposed from the 3P-DGAC method. Solid curves indicate positive values, and dashed curves indicate negative values. Stippling indicates regions where at least 10 of 13 simulations in the model group agree on the sign of the ensemble mean. (b) Changes in annual mean intensities (unit: $10^{10}$ kg/s) of meridional circulation in the NH and SH. (c) The latitudes of the center of annual mean precipitation between 20°S and 20°N (unit: °). (d) Hemispheric asymmetry (NH minus SH) of energy flux into the atmosphere (unit: W·m$^{-2}$). (e) Changes in the integrated atmospheric meridional heat transport across the equator (unit: PW).

[Figure]

Figure 11: (a) Changes in ZMS (shading, unit: $10^{10}$ kg s$^{-1}$) averaged between 10°S/N for the mid-Pliocene with respect to the PI simulation, overlaid by the climate mean ZMS for the PI

simulation (contours). The zonal wind $\vec{V}_Z$ is used to calculate ZMS, which is decomposed from the 3P-DGAC method. The contours represent the climate mean ZMS for the PI simulation. Solid curves indicate a positive value, and dashed curves show a negative value. Stippling indicates regions where at least 10 of 13 simulations in the model group agree on the sign of the ensemble mean. (b) is the vertical integrated ZMS in (a). The gray and pink shading indicates 1 standard deviation of individual model departure from the MMM mean of MSF for the PI and mid-Pliocene simulations, respectively.

**16. I think stippling should be used on the non-significant regions so that we have a clear view of the significant results.**
**Response:** Thanks for your comment. We have changed the stippling to be smaller in the significant regions to try to make it more clear in the new version.

---

## Author Comment (AC3)

**Author response to Reviewer #1**

**The authors thank the reviewers and editors for beneficial and helpful suggestions for this manuscript. We have carefully revised the manuscript according to the suggestions. According to the reviewer's comments, the following replies have been made in blue.**
* * *
**Reviewer's Comments to Authors:**

This paper evaluates the contribution of thermodynamical and dynamical effects to changes in P minus E in the simulated Pliocene warming. It decomposes the moisture budget into thermodynamical and dynamical contribution, which are further decomposed by applying the 3-pattern decomposition of the atmospheric circulation. The authors find that both mechanisms have significant contributions in explaining changes in PmE. In general, they find that thermodynamical effect can explain the wetter tropics and drier subtropics, however, when looking to more regional changes within the tropics and subtropics, dynamical effects have an important contribution. The paper goes on to explore possible reasons for the changes in the moisture budget components.

I think that the methods applied to achieve the main objective are appropriate and that the paper shows an interesting contribution to the research in the field. Therefore, I recommend the publication of this manuscript by the journal Climate of the Past, but I see some major issues that need to be addressed before publication.

**General comments:**

1. The main objective of the is study is to evaluate the relative contribution of thermodynamical and dynamical effects in the PlioMIP models. However, the paper does not compare the relative contributions of the two. The authors correctly applied the moisture budget decomposition, but they do not quantitively compare how much of each of the components contributes to the total PmE pattern. I think this analysis is crucial to this evaluation and would greatly improve the manuscript.

**Response:** Thanks for the comments. To quantitively compare the contribution of the dynamic and thermodynamic components to the total PmE change, we firstly calculate the zonal mean of moisture budget components as shown in supplement Figure 1a. The changes in PmE generally show a wet-regions-getting-wetter and dry-regions-getting-drier pattern during the mid-Pliocene, and this is mainly contributed by the thermodynamic term. Furthermore, the thermodynamic term leads to increase in PmE by ~58.6% over tropic region, and decrease in PmE by ~84.6% over subtropics, respectively. The dynamic effect has relative smaller contribution, with increase in PmE by ~7.9% in tropic and ~10.5% in subtropics. In fact, the $\delta TH$ component does not describe the full contribution of the changes in PmE everywhere. For example, over the North Indian Ocean,the $\delta MCD_{D\_Z}$ (decomposed by the $\delta MCD$ term) is the first-order contribution to strengthen the dynamic effect (by ~45%) and hence enhance the PmE over north Indian ocean (Figure 4g), which is caused by the shift of Walker circulation (Figure 10), the contribution of thermodynamic is negligible. We show the contribution of each moisture budget components over north Indian ocean in supplementary Figure 2.

**We have added related sentences in the new version as follows:**

"In general, the thermodynamic term increases PmE by ~58.6% over tropic region, and decrease in PmE by ~84.6% over subtropics (not shown), respectively." (See Line 264-265)

"..., the convergence of zonal circulation anomalies ($\delta MCD_{D\_Z}$) is the first-order contribution to strengthen the dynamic effect (by ~45%) and hence enhance the PmE (Fig.4(g))." (See Line 312-313)

[Figure]

Figure S1. (a) The zonal average of the moisture budget components changes for PlioMIP2 models. (b) The average of moisture budget components in tropics (5°S-15°N) and subtropics (30°-5°S and 15°-30°N). Units: mm/day.

[Figure]

Figure S2. The average of moisture budget components over north Indian ocean (0-20°N, 40°-100°E). unit: mm/day.

2. The authors have produced interesting results that should be introduced more carefully. There is a contradiction in the abstract. The authors initially exemplify the changes in PmE over the Pacific ITCZ as a result of the thermodynamical effect. And a few lines below, they evaluate that the ITCZ shifts 1° northward (which is a result of dynamical changes). To me, the results show that in a very general (large-scale) manner the thermodynamical effects explain the wetter deep tropics and drier subtropics. But when looking to regional changes within the tropics and subtropics the

dynamical effect plays an important role (i.e. northward ITCZ shift, southward SPCZ shift, wetter north Indian ocean).

**Response:** The reviewer's suggestions are very beneficial. The reviewer's opinion is right. Indeed, our results indicate that the thermodynamical effects explain the wetter deep tropics and drier subtropics in a large-scale manner. These results can also be found in Figure S1 (see comment 1). Although the dynamic term is relative smaller than thermodynamic term in a general manner (Figure S1), the dynamic effects can not be neglect when looking to regional changes in the low latitudes. **To avoid the confusion in abstract, we have revised and added the related sentences as follows:**

"Note that the dynamic effect play a more important role in regional PmE changes (i.e., northward ITCZ shift and wetter in north Indian ocean)." (See Line 44-45).

3. It was not clear to me what boundary conditions the PlioMIP2 models have applied. For example, did they apply 400 ppm $CO_2$? Are there changes in the extension of the ice sheets in both hemispheres? Vegetation? Changes in the configuration of the continents? Also, I think there should have a final discussion as how these boundary conditions could have affected the simulated PmE changes.

**Response:** Thanks for the comments. We use a suite of model simulations from 13 the Pliocene Model Intercomparison Project Phase 2 (PlioMIP2) participating the Coupled Model Intercomparison Project phase 6 (CMIP6) (Haywood et al., 2016). Boundary conditions for mid-Pliocene simulations are derived from the latest iteration of the U.S. Geological Survey PRISM data set (PRISM4) (Dowsett et al., 2016), including land versus sea, vegetation, soils, lakes, land ice cover, topography and bathymetry. The concentration of atmospheric $CO_2$ level for mid-Pliocene and PI simulations are prescribed at 400ppm and 280ppm, respectively. Other solar output, trace gases and orbital parameters for mid-Pliocene simulations are set to be consistent with each model's PI simulation. Modeling groups are given the option to either prescribe vegetation changes or simulate vegetation changes using a dynamic global vegetation model. And only one modeling group (COSMOS) in the suite of simulations we analyzed opted to use a dynamic configuration for vegetation.

We agree with the reviewer that it is important to discuss the impact of boundary conditions on changes in PmE during mid-Pliocene. Note that the large-scale atmospheric circulation changes are related to the temperature pattern anomalies. Earlier studies demonstrate that the temperature changes during mid-Pliocene are controlled by the combined effects of external forcings, including the atmospheric $CO_2$ concentration, land cover (surface vegetation and land ice cover), and topography (Haywood et al., 2016). Any changes in each boundary conditions could lead to changes in PmE through dynamic (i.e., atmospheric circulation changes) or thermodynamic (i.e., atmospheric moisture changes) effects. For example, several studies highlight the role of remote biophysical effects in the northern mid-high latitudes in driving the variation of monsoon rainfall in low latitudes, which could lead to land surface characteristics changes (i.e., evapotranspiration and albedo),

altering shifts of ITCZ due to the imbalance of atmospheric energy budget and hence changing monsoon rainfall (Swann et al., 2014; Chase et al., 2000; Mahmood et al., 2014). Proxy and simulations suggest that the needleleaf tree shifts greatly and expands northward in eastern Eurasia during mid-Pliocene (Zhang and Jiang, 2017; Feng et al., 2021). These changes in land cover during mid-Pliocene may alter PmE in the low latitudes as well, especially in monsoon regions. In addition, some studies indicate that there exist uncertainties of boundary conditions of changing South Asian summer monsoon hydrological cycling. Sarathchandraprasad et al. (2017) indicate the tectonically induced reorganization of the Indonesian Throughflow can strengthen the South Asian summer monsoon (SASM) during mid-Pliocene due to the increased cross-equatorial pressure gradient. Recent study by Prescott et al., (2019) highlight the substantially influence of orbital forcing on the changes in SASM during mid-Pliocene. The simulations suggest that tectonic uplifts in South African Plateaus can strengthen the SASM as well (Zhang and Liu, et al., 2013). Based on these studies, the boundary conditions applied by the PlioMIP2 models are important to hydrological cycling during mid-Pliocene in low latitudes. However, since not all models carry out the sensitivity experiments designed in PlioMIP2, it remains difficult to distinguish which change in boundary conditions is more dominant for the PmE changes in low latitudes.

**We have added the related description of boundary conditions and discussion of the possible effects in the new version as follows:**

"..., including soils, lakes, land ice cover, vegetation, topography, bathymetry. The $CO_2$ level for mid-Pliocene and PI simulations are set at 400ppm and 280ppm, respectively." (See line 104-106)

"Zhang et al. (2016) indicate that the combined influence of SST and $CO_2$ level, as well as the vegetation changes, play very important role in changing the atmospheric circulation over North Africa during mid-Pliocene, owing to the increased net atmospheric energy there." (See line 214-216)

"Note that the global temperature during mid-Pliocene is controlled by the combined effects of boundary conditions (e.g., CO2 level, vegetation and topography) (Haywood et al., 2016). Any changes in each boundary condition could induce large-scale hydrological cycling changes. For example, the role of remote biophysical effects in the northern mid-high latitudes is highlighted in driving the variation of monsoon rainfall in low latitudes and shift of ITCZ, since the needleleaf tree expands greatly northward in eastern Eurasia during mid-Pliocene (Chase et al., 2000; Swann et al., 2014; Mahmood et al., 2014; Zhang and Jiang, 2014; Burls and Fedorov, 2017). In addition, some studies indicate that there exist uncertainties of boundary conditions of changing South Asian summer monsoon hydrological cycling. Sarathchandraprasad et al. (2017) indicate the tectonically induced reorganization of the Indonesian Throughflow can strengthen the SASM during mid-Pliocene due to the increased cross-equatorial pressure gradient. Recent study by Prescott et al. (2019) highlight the substantially influence of orbital forcing on the changes in SASM during mid-Pliocene. The simulations suggest that tectonic uplifts in South African Plateaus can strengthen the SASM as well (Zhang and Liu, 2013). Based on these studies, the

boundary conditions applied by the PlioMIP2 models are important to hydrological cycling during mid-Pliocene in low latitudes. However, the relative impact of boundary conditions on hydrological cycling still remains uncertainties. In addition, not all models carry out the sensitivity experiments designed in PlioMIP2, increasing the difficulty of exploring their relative contributions to PmE changes, and these questions need to be further explored in the future." (See Line 529-543)

**References:**

Haywood, A. M., Dowsett, H. J., Dolan, A. M., Rowley, D., Abe-Ouchi, A., Otto-Bliesner, B., Chandler, M. A., Hunter, S. J., Lunt, D. J., Pound, M., and Salzmann, U.: The Pliocene Model Intercomparison Project (PlioMIP) phase 2: scientific objectives and experimental design, Clim. Past, 12, 663-675, https://doi.org/10.5194/cp-12-663-2016, 2016.

Zhang R. and Jiang D.. 2014. Impact of vegetation feedback on the mid-Pliocene warm climate. Advances in Atmospheric Sciences, 31: 1407-1416.

Swann, A.L., Fung, I.Y., Liu, Y. and Chiang, J.C., 2014. Remote vegetation feedbacks and the mid-Holocene green Sahara. Journal of climate, 27(13), pp.4857-4870.

Chase, T.N., Pielke Sr, R.A., Kittel, T.G.F., Nemani, R.R. and Running, S.W., 2000. Simulated impacts of historical land cover changes on global climate in northern winter. Climate dynamics, 16(2), pp.93-105.

Mahmood, R., Pielke Sr, R.A., Hubbard, K.G., Niyogi, D., Dirmeyer, P.A., McAlpine, C., Carleton, A.M., Hale, R., Gameda, S., Beltrán-Przekurat, A. and Baker, B., 2014. Land cover changes and their biogeophysical effects on climate. International journal of climatology, 34(4), pp.929-953.

Burls, N. J. and Fedorov, A. V.: Wetter subtropics in a warmer world: contrasting past and future hydrological cycles, Proc. Natl. Acad. Sci. U. S. A., 114, 12888, https://doi.org/10.1073/pnas.1703421114, 2017.

Sarathchandraprasad, T., Tiwari, M. and Behera, P., 2021. South Asian Summer Monsoon precipitation variability during late Pliocene: Role of Indonesian Throughflow. Palaeogeography, Palaeoclimatology, Palaeoecology, 574, p.110447.

Zhang R. and Liu X.D., 2010. The effects of tectonic uplift on the evolution of Asian summer monsoon climate since Pliocene. Chinese Journal of Geophysics, 53(6), pp.948-960.

4. Section 3 is an important section of paper that introduces the simulated PmE changes that will be further explored in the next sections and should be carefully revised. To me the key message of section 3 is that, in a first look to changes in PmE, there seem to have important contributions from both TH and MCD, but the authors mainly discuss TH (see comment 2 and detailed comments below).

**Response:** Thanks for the comments. We have added a few sentences to try to avoid this confusion as follows:

"Note that there is moistening signal in SPCZ's southern part in the tropical southern Pacific." (See line 207-208)

"Earlier studies indicate these features of changes in PmE at low latitudes are linked to the increased specific humidity (i.e., changes in thermodynamic effect). However, there are some opposite phenomenon as well, when looking to the regional changes in PmE (i.e., north Inidan Ocean, North Africa and SPCZ). These may suggest that another factor, such as atmospheric circulation anomalies (i.e., changes in dynamic effect), may play an important role in changing regional PmE pattern at low latitudes." (See Line 229-233)

5. Section 5.1: it is interesting to examine the global pattern in wind divergence and humidity. But SST pattern must be analysed more thoroughly, especially in the tropics where local SST will probably help to explain the thermodynamical effect.
Xie, S.-P., C. Deser, G. A. Vecchi, J. Ma, H. Teng, and A. T. Wittenberg, 2010: Global warming pattern formation: Sea surface temperature and rainfall. J. Climate, 23, 966–986, https://doi.org/10.1175/2009jcli3329.1.

**Response:** The reviewer's suggestions are very beneficial. Figure S3 shows the changes in annual mean SST and wind at 850 hPa. The SST warming is amplified in the northwest tropical Indian Ocean, whereas it is reduced off the Indonesian coast, showing tropical Indian Ocean dipole (IOD)-like pattern. The sharp SST gradients drive strong southeasterly wind anomalies (Figure S3) over the equator. Figure S4 show the changes in annual mean of zonal wind (U) versus changes in SST for 13 PlioMIP2 models. These results can also be seen in Figure S4. As expected, the models with an enhanced zonal SST gradient across the equatorial Indian Ocean tend to produce increased easterly wind anomalies, with the inter-model correlations of -0.9. Xie et al. (2010) suggest that this easterly wind anomalies may shoal the thermocline in the east, helping cool SST there via upwelling. This indicates that this SST anomalies over tropical Indian Ocean may be related to the Bjerknes feedback. Studies indicate that the simulated North Atlantic warming is related to an intensified mid-Pliocene AMOC (Li et al., 2020). However, Zhang et al. (2021) suggest that the increased background ocean vertical mixing parameters can be also responsible for the warm SSTs there. Note that the relative cool SST in Southeast Pacific and Atlantic, which is collocated with the intensified southeast trades, suggesting the role of WES feedback (Xie et al., 2010).

According to the comments, we have added more discussion of the SST pattern as follows:

"Note that the SST warming is amplified in the northwest tropical Indian Ocean, whereas it is reduced off the Indonesian coast, showing tropical Indian Ocean dipole (IOD)-like pattern. The sharp SST gradients drive strong southeasterly wind anomalies on the equator (not shown). Xie et al. (2010) suggest that this easterly wind anomaly may shoal the thermocline in the east, helping cool SST there via upwelling, and indicating this SST anomaly over tropical Indian Ocean may be related to the Bjerknes feedback. The simulated North Atlantic warming might be related to an intensified mid-Pliocene AMOC (Li et al., 2020). However, Zhang et al. (2021) suggest that the increased background ocean vertical mixing parameters could also contribute to the warm SSTs there. In addition, the relative smaller SST warming in

Southeast Pacific and Atlantic, which is collocated with the intensified southeast trades, suggests the role of wind-evaporation-SST feedback (Xie et al., 2010). These SST warming patterns are consistent with current studies (Haywood et al., 2020; Williams et al., 2021)." (See line 360-371)

References:

Xie, S.-P., C. Deser, G. A. Vecchi, J. Ma, H. Teng, and A. T. Wittenberg, 2010: Global warming pattern formation: Sea surface temperature and rainfall. J. Climate, 23, 966–986, https://doi.org/10.1175/2009jcli3329.1.

Li, X., Guo, C., Zhang, Z., Otterå, O.H. and Zhang, R., 2020. PlioMIP2 simulations with NorESM-L and NorESM1-F. Climate of the Past, 16(1), pp.183-197.

Zhang, Z., Li, X., Guo, C., Otterå, O.H., Nisancioglu, K.H., Tan, N., Contoux, C., Ramstein, G., Feng, R., Otto-Bliesner, B.L. and Brady, E., 2021. Mid-Pliocene Atlantic Meridional Overturning Circulation simulated in PlioMIP2. Climate of the Past, 17(1), pp.529-543.

[Figure]

Figure S3. Change in multimodel mean SST (shading, unit: ℃) and wind at 850hPa (vectors, unit: m/s).

[Figure]

Figure S4. Intermodel scatterplots of changes in west-minus-east SST difference (west box:10°S-10°N,50°-70°E; east box: 0°-10°S, 90°-110°E) vs surface zonal wind in central equatorial Indian Ocean (CEIO; 70°-90°E).

**Specific comments:**
6. Line 54: Changes in the hydrological cycle are a response to regional and global climate anomalies. I would rephrase to "Global warming can induce regional and global anomalies in the Earth's hydrological cycle…" or similar.
**Response:** Thanks for the comments. We have changed this sentence as "Global warming can induce regional and global anomalies in the Earth's hydrological cycle…". (See Line 55)

7. Line 203: Figure 1a indicates that changes over the Maritime Continent are not consistent across models. I suggest removing mentions to Maritime Continent and rather focus on the North Indian Ocean, where changes are significant.
**Response:** Thanks for the comments. We have removed the "Maritime Continent". (See Line 206)

8. Line 207: When discussing changes in Southeast Asia and Sahel the authors must discuss the effect of vegetation. Do the PlioMIP models apply changes in vegetation?
**Response:** Thanks for the comments. As we discussed in comment 3, boundary conditions for MP simulations are derived from the latest iteration of the U.S. the vegetation is from Geological Survey PRISM data set (PRISM4) (Dowsett et al., 2016). Modeling groups were given the option to either prescribe vegetation changes or simulate vegetation changes using a dynamic global vegetation model. For the latter experiment, model simulations were started with pre-industrial vegetation and the model was allowed to spin up until a new equilibrium distribution of vegetation was achieved. Note that only one modeling group (COSMOS) in the suite of simulations we analyzed opted to use a dynamic configuration for vegetation.

**According to the reviewer's suggestion, we have added the related discussion in the new version as follows:**

"..., including soils, lakes, land ice cover, vegetation, topography, bathymetry. The $CO_2$ level for mid-Pliocene and PI simulations are set at 400ppm and 280ppm, respectively." (See Line 104-105)

"Zhang et al. (2016) indicate that the combined influence of SST and $CO_2$ level, as well as the vegetation changes, play a very important role in changing the atmospheric circulation over North Africa during mid-Pliocene, owing to the increased net atmospheric energy there. Additionally, the expansion of vegetation into the Sahara region tend to decrease the surface albedo, which can enhance the Saharan Heat Low and hence impact rainfall over West Africa, reflecting the vegetation-albedo feedback (Charney, 1975). Recent studies indicate the enhanced vegetation in PlioMIP2 ensemble is likely to have contributed to the increasing of the mid-Pliocene West African summer rainfall (Haywood et al., 2020; Berntell et al., 2021)." (See Line 214-220)

"Note that the global temperature during mid-Pliocene is controlled by the combined effects of boundary conditions (e.g., CO2 level, vegetation and topography) (Haywood et al., 2016). Any changes in each boundary condition could induce large-scale hydrological cycling changes. For example, the role of remote biophysical effects in the northern mid-high latitudes is highlighted in driving the variation of monsoon rainfall in low latitudes and shift of ITCZ, since the needleleaf tree expands greatly northward in eastern Eurasia during mid-Pliocene (Chase et al., 2000; Swann et al., 2014;  Mahmood et al., 2014; Zhang and Jiang, 2014; Burls and Fedorov, 2017)." (See Line 529-534)

References:
Zhang, R., Zhang, Z., D Jiang, Yan, Q., Zhou, X., and Cheng, Z.. 2016. Strengthened african summer monsoon in the mid-piacenzian. Advances in Atmospheric Sciences.
Swann, A.L., Fung, I.Y., Liu, Y. and Chiang, J.C., 2014. Remote vegetation feedbacks and the mid-Holocene green Sahara. Journal of climate, 27(13), pp.4857-4870.
Chase, T.N., Pielke Sr, R.A., Kittel, T.G.F., Nemani, R.R. and Running, S.W., 2000. Simulated impacts of historical land cover changes on global climate in northern winter. Climate dynamics, 16(2), pp.93-105.
Mahmood, R., Pielke Sr, R.A., Hubbard, K.G., Niyogi, D., Dirmeyer, P.A., McAlpine, C., Carleton, A.M., Hale, R., Gameda, S., Beltrán-Przekurat, A. and Baker, B., 2014. Land cover changes and their biogeophysical effects on climate. International journal of climatology, 34(4), pp.929-953.
Zhang R. and Jiang D.. 2014. Impact of vegetation feedback on the mid-Pliocene warm climate. Advances in Atmospheric Sciences, 31: 1407-1416.
Burls, N. J. and Fedorov, A. V.: Wetter subtropics in a warmer world: contrasting past and future hydrological cycles, Proc. Natl. Acad. Sci. U. S. A., 114, 12888, https://doi.org/10.1073/pnas.1703421114, 2017.
Berntell, E., Zhang, Q., Li, Q., Haywood, A.M., Tindall, J.C., Hunter, S.J., Zhang, Z., Li, X., Guo, C., Nisancioglu, K.H. and Stepanek, C., 2021. Mid-Pliocene West African Monsoon rainfall as simulated in the PlioMIP2 ensemble. Climate of the Past, 17(4), pp.1777-1794.
Charney, J. G.: Dynamics of deserts and drought in the Sahel, Q. J. Roy. Meteor. Soc., 101, 193–202, https://doi.org/10.1002/qj.49710142802, 1975.
Haywood, A. M., Tindall, J. C., Dowsett, H. J., Dolan, A. M., Foley, K. M., Hunter, S. J., Hill, D. J., Chan, W. L., Abe-Ouchi, A., Stepanek, C., Lohmann, G., Chandan, D., Peltier, W. R., Tan, N., Contoux, C., Ramstein, G., Li, X., Zhang, Z., Guo, C., Nisancioglu, K. H., Zhang, Q., Li, Q., Kamae, Y., Chandler, M. A., Sohl, L. E., Otto-Bliesner, B. L., Feng, R., Brady, E. C., Von der Heydt, A. S., Baatsen, M. L. J., and Lunt, D. J.: The Pliocene model intercomparison project phase 2: large-scale climate features and climate sensitivity, Clim. Past, 16, 2095–2123, https://doi.org/10.5194/cp-16-2095-2020, 2020.

9. Line 213: It is not clear to me why a drier extratropical North Atlantic is related to increased SST. Wouldn't an increased SST drive increased precipitation (note increased humidity in the north Atlantic in Figure 6b)? Also, Figure 5a does not show SST. Do you mean Figure 6a? I suggest leaving this discussion to Section 5.1 to avoid referencing figure 5 without mentions to figure 3 and 4.

**Response:** The reviewer's comments are right. In fact, we have discussed the cause of drier extratropical North Atlantic in Section 5.1. According your comments, we remove the related misunderstanding description as follows:

We have removed "..., and this change might be caused by the simulated largely increased SST in the North Atlantic (Fig. 5(b)) ". (See in Line 222)

10. Line 215: I suggest a reformulation of this paragraph to include that both TH and MCD seems to be important for the PmE changes, as it shown later in the manuscript.

**Response:** Thanks for you comments. We have added some sentences to highlight the importance of δTH and δMCD in this paragraph as follows:

"Earlier studies indicate these features of changes in PmE at low latitudes are linked to the increased specific humidity (i.e., changes in thermodynamic effect). However, there are some opposite phenomenon as well, when looking to the regional changes in PmE (i.e., north Inidan Ocean, North Africa and SPCZ). These may suggest that another factor, such as atmospheric circulation anomalies (i.e., changes in dynamic effect), may play an important role in changing regional PmE pattern at low latitudes." (See Line 229-233)

11. Figure 1b: Why the authors choose to show the models' standard deviation? To be consistent with your choice of showing significance based on model agreement in Figure 1a, it would be more appropriate to show interquartile range (non-parametric statistics).

**Response:** Thanks for you comments. Following your suggestion, we have changed the model's standard deviation to its interquartile range in Figure 1b as follows.

[Figure]

Figure 1: (a) Changes in multimodel mean (MMM) PmE for the mid-Pliocene compared with the PI simulation (shading), overlaid by the climatological MMM PmE of the PI simulation (for the contours, a solid line indicates positive values and a dashed line indicates negative values). The red solid curves represent the zero value. (b) The zonal average of the change in PmE, where the shading indicates the interquatile range among models. Stippling (left) indicates regions where at

least 10 of 12 simulations in the model group agree on the sign of the ensemble mean. Units: mm·day$^{-1}$.

12. Line 396: Figure 9a indicates that the intensification of the PWC is not significant. I wouldn't mention the value.
**Response:** Thanks for the comments. We have removed the sentence "with the mean PWC intensity strengthened by $1\times10^{10}$ kg/m$^2$ compared with the PI simulation". (See Line 430)

13. Lines 395-405: I think that the key message of this paragraph is that the change in the intensity of the PWC does not explain the change in PmE in the Indian Ocean. As such, here it should be clearer for the reader that the changes in PWC intensity, dSST and dSLP are not consistent (nor significant) across models. Nonetheless, changes in dSST and dSLP can explain changes in PWC intensity. This paragraph needs to be reformulated.
**Response:** Thanks for the comments. Figure S5 further examines the relationships between the change in dSST versus the changes in dSLP and PWC intensity among 13 PlioMIP2 models. Indeed, the change in PWC intensity is highly related to the changes in zonal SST and SLP gradient. Models with an enhanced zonal SST gradient across the equatorial Indo-Pacific tend to produce weaker zonal SLP gradient and decreased PWC, with the inter-model correlations of -0.95 and -0.75, respectively. We have added the related sentences to avoid the confusion as follows:

"As expected, the models with an enhanced zonal SST gradient across the equatorial Indo-Pacific tend to produce weaker zonal SLP gradient and decreased PWC (Figure not shown), with the inter-model correlations of -0.95 and -0.75, respectively. Note that, ..." (See Line 448-450)

[Figure]

Figure S5. Changes in annual mean of the (a) intensities of Pacific Walker circulation (PWC; unit: $10^{10}$ kg/s) and (b) dSLP (units: hPa) versus changes in dSST (units: K) for 13 PlioMIP2 models. The inter-model correlation (r) is shown in each panel. Here, the PWC intensity is defined as the vertically integrated ZMS (Bayr et al., 2014; Schwendike et al., 2014) averaged in the equatorial Pacific (140°E-120°W). The dSLP and dSST are defined as the difference in SLP and SST across the equatorial Indo-Pacific (160°W-80°W, 5°S-5°N minus 80°E-160°E, 5°S-5°N).

14. Figure 9: Panel 'b' seems lost in this Figure. I would consider moving it to Figure This way, all results explaining the shift in the zonal circulation would be clearly shown in one figure. There is no mention to Figure 9b in between lines 395-405.

**References:** Thanks for the comments. Actually, we mention the description of Figure 9b in between lines 449-457, and mainly discuss the movement of the Pacific Walker circulation. So we decide to keep it in Figure 9.

15. Line 422: Do you mean a westward expansion of the PWC and a westward displacement over the Indian Ocean?

**References:** The review's question is right. We have revised the confusion sentence as follows:

"..., indicating a westward expansion of the PWC (Fig. 11(b))." (See Line 472)

16. Line 437: westward 'expansion' of the PWC?

**References:** The reviewer's question is right. We have revised the confusion sentence as follows:

"..., resulting from the westward expansion of the PWC" (See line 471)

17. Section 6: the authors must discuss the main results on the light of the boundary conditions applied by the PlioMIP models and their uncertainties, to guide future research.

**References:** Thanks for your comment. We have discussed the impact of boundary conditions on hydrological cycling in section6 as mentioned above in comment 3.

**Technical corrections:**

18. Line 68: Redundance. "enhance the increase in atmospheric…".

**Response:** Thanks for your carefully reading. We have removed the word "enhance the" in this sentence (See line 68).

19. Figure 1a: I recommend changing the colour of the stipplings. It is hard to see even when zooming.

**Response:** Thanks for your comment. We have changed the color of the stipplings in the Figure 1a from gray to black. See comment 11.

20. Line 364: include a reference to Figure 8 for clarity.

**Response:** Thanks for your comment. We have added the "Fig. 9" after the sentence "... winds are located in the central Pacific". (See line 392)

21. Figure 10a: indicate the intensity of the PI contours.

**Response:** Thanks for your comment. We have added the intensity of the PI contours in Figure 11 as follows:

[Figure]

Figure 11: (a) Changes in ZMS (shading, unit: $10^{10}$ kg s$^{-1}$) averaged between 10°S/N for the mid-Pliocene with respect to the PI simulation, overlaid by the climate mean ZMS for the PI simulation (contours). The zonal wind $\vec{V}_Z$ is used to calculate ZMS, which is decomposed from the 3P-DGAC method. The contours represent the climate mean ZMS for the PI simulation. Solid curves indicate a positive value, and dashed curves show a negative value. Stippling indicates regions where at least 10 of 13 simulations in the model group agree on the sign of the ensemble mean. (b) is the vertical integrated ZMS in (a). The gray and pink shading indicates 1 standard deviation of individual models departure from the MMM mean of MSF for the PI and mid-Pliocene simulations, respectively.

---

## Author Response (AR2)

The authors thank the reviewers and editors for beneficial and helpful suggestions for this manuscript. We have carefully revised the manuscript according to the suggestions. According to the comments, the revision in manuscript have been made in **red**.
* * *
1. We have changed "the thermodynamic effect" to "due to the thermodynamic effect" in line 42.

2. We have changed "a more important role" to "a more important role than the thermodynamic effect" in line 44.

3. We have changed "PmE (precipitation minus evaporation)" to "precipitation minus evaporation (PmE)" in line 45.

4. We have changed "rainfall" to "precipitation" in line 62.

5. We have changed "focus" to "focused" in line 63.

6. We have added "the" in line 63, 248, 250, and 252.

7. We have moved the sentences "In PlioMIP2 models, the boundary conditions have been updated using the new version of the U.S. Geological Survey PRISM4 dataset (Dowsett et al., 2016; Haywood et al., 2016), including soils, lakes, land ice cover, vegetation, topography, and bathymetry. The CO2 level for mid-Pliocene and PI simulations are set at 400ppmv and 280ppmv, respectively" to the section2.

8. We have changed "SPCZ's southern part in the tropical southern Pacific." to "the southern part of the SPCZ in the tropical southern Pacific" in line 210.

9. We have changed "increase" to "increased" in line 221.

10. We have changed "hydrology" to "hydrological changes" in line 244.

11. We have changed "impacting" to "adding uncertainties to" in line 257.

12. We have changed "reduce" and "increase" to "reduced" and "increased", respectively, in line 310.

13. We have changed "cooling" to "lower" in line 366.